# Guiding Cross-Modal Representations with MLLM Priors via Preference Alignment

**Pengfei Zhao, Rongbo Luan, Wei Zhang, Peng Wu, Sifeng He**[*]
Apple
{pzhao23, he_sifeng}@apple.com

## Abstract

Despite Contrastive Language–Image Pre-training (CLIP)'s remarkable capability to retrieve content across modalities, a substantial modality gap persists in its feature space. Intriguingly, we discover that off-the-shelf MLLMs (Multimodal Large Language Models) demonstrate powerful inherent modality alignment properties. While recent MLLM-based retrievers with unified architectures partially mitigate this gap, their reliance on coarse modality alignment mechanisms fundamentally limits their potential. In this work, We introduce MAPLE (Modality-Aligned Preference Learning for Embeddings), a novel framework that leverages the fine-grained alignment priors inherent in MLLM to guide cross-modal representation learning. MAPLE formulates the learning process as reinforcement learning with two key components: (1) Automatic preference data construction using off-the-shelf MLLM, and (2) a new Relative Preference Alignment (RPA) loss, which adapts Direct Preference Optimization (DPO) to the embedding learning setting. Experimental results show that our preference-guided alignment achieves substantial gains in fine-grained cross-modal retrieval, underscoring its effectiveness in handling nuanced semantic distinctions.

## 1 Introduction

Cross-modal retrieval, which aims to retrieve relevant content across different modalities (e.g., retrieving images with text queries), has long been a core topic in the vision-language research community. It also serves as a fundamental building block for a wide range of downstream task, including visual question answering, retrieval-augmented generation, and increasingly, LLM-based multi-modal agent systems. Despite the remarkable success of large-scale contrastive pretraining frameworks such as CLIP [1], a fundamental challenge still remains: the *modality gap*, i.e., the discrepancy in feature representation between visual and textual modalities. This modality gap often manifests as a spatial separation between image and text embeddings in the shared latent space, and also limits the effectiveness of learned representations in various retrieval tasks [2–4]. Prior works [2, 4] have shown that this misalignment might be attributed from architectural choices, training dynamics, and even input information imbalance.

While prior works mainly focused on measuring the modality gap using explicit embeddings [2, 3], the alignment properties of models that operate directly via logits, such as off-the-shelf Multimodal Large Language Models (MLLMs), remain less explored. To better understand the modality gap across different model architectures, we propose a unified metric based on the 1-Wasserstein Distance (WD) [5] that enables direct comparison between logit-based and embedding-based models. For MLLMs, we compute pairwise alignment scores using output logits (as detailed in Section 2), while for embedding models, we use cosine similarities. In both cases, we apply WD to measure the discrepancy between the similarities distributions. Intriguingly, through this quantitative comparison,

---

[*]Corresponding Author, Project Lead

39th Conference on Neural Information Processing Systems (NeurIPS 2025).

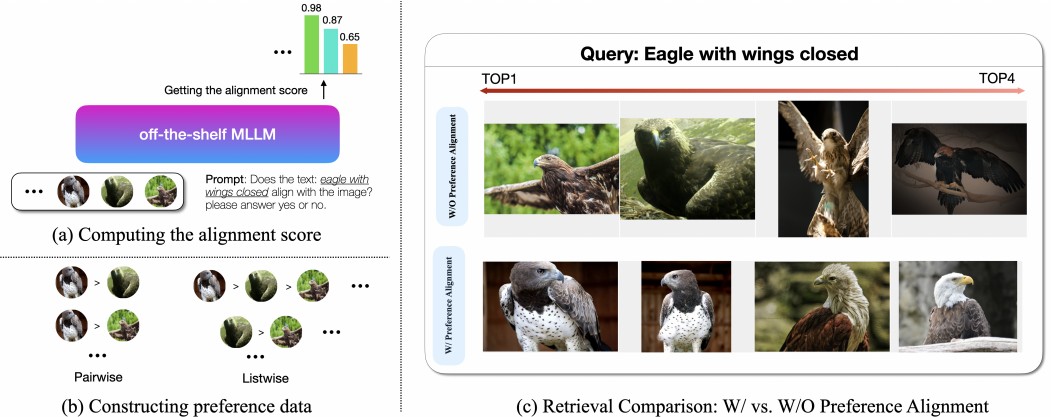

(a) Computing the alignment score

(b) Constructing preference data

(c) Retrieval Comparison: W/ vs. W/O Preference Alignment

Figure 1: (a) **Computing the Alignment Score**: Prompting the off-the-shelf MLLM to output "yes" or "no" token for the paired text-image, and then calculating the alignment score based on "yes" and "no" token logits. (b) **Constructing Preference Data**: Constructing pairwise or listwise preference data based on the calculated alignment scores. (c) **Retrieval Comparison**: Through preference alignment, the retrieval model can capture fine-grained distinctions between retrieved images under the same query.

we discover that MLLMs like Qwen2-VL [6, 7] exhibit strong inherent modality alignment capabilities even without relying on explicit embedding representations. Our analysis, illustrated in Appendix Figure 3, reveals that MLLMs demonstrate an emergent ability to align image and text inputs more effectively than CLIP.

Recent efforts have fine-tuned MLLMs for retrieval by enabling cross-modal representation learning [8–10]. However, transitioning from generative architectures to MLLM-based retrievers often diminishes the inherent alignment capabilities of MLLMs, even after further fine-tuning. Motivated by the above observation, we aim to adapt MLLM for retrieval tasks while preserving their strong cross-modal alignment strength.

To this end, we propose **MAPLE** (Modality-Aligned Preference Learning for Embeddings), a novel framework that bridges the alignment capabilities of off-the-shelf MLLM with MLLM-based retrieval model. MAPLE transfers MLLM's alignment capabilities to embedding spaces through two key dimensions: data-level preference construction and training-strategy-level preference alignment. At the data level, we retrieve top-K hard samples for each anchor and leverage MLLM to score their matching degree with the anchor's corresponding cross-modal content. These alignment scores are then used to construct both pairwise preferences and listwise preferences for training. At the training level, we derive the Relative Preference Alignment (RPA) loss from Direct Preference Optimization (DPO) [11], specifically adapted for embedding models to achieve fine-grained cross-modal alignment. Through optimization with the RPA loss on these constructed preferences, as illustrated in Figure 1 c, our method effectively captures subtle distinctions between retrieved images under the same query.

The related work is in Appendix D. The main contributions of our paper include:

- We propose Wasserstein distance as a unified metric to measure the modality gap for both logit-based and embedding-based models, revealing that off-the-shelf MLLMs inherently exhibit strong cross-modality alignment capabilities.

- We introduce the MAPLE framework for extracting powerful multimodal embeddings. Our approach features a novel strategy that automatically leverages the inherent modality alignment capabilities of MLLM to construct preferences after hard sample mining. Additionally, we explore the adaptation of Direct Preference Optimization (DPO) for cross-modality representation learning, which yields substantial improvements in fine-grained retrieval.

- We validate our framework through comprehensive experiments on different benchmarks including general retrieval (e.g., COCO [12], Flickr30K [13]), fine-grained retrieval (e.g. Winoground [14], NaturalBench [15], MMVP [16], BiVLC [17]). The experimental results

demonstrate the superiority and effectiveness of our approach in both general and fine-grained retrieval tasks.

## 2 Preliminaries and Notation

Given a batch of $N$ image-text pairs $\{(x_i^{\text{img}}, x_i^{\text{txt}})\}_{i=1}^N$, unlike CLIP, which utilizes separate encoders to extract embeddings for each modality, we leverage the MLLM, a unified architecture, to obtain embeddings for each modality. Following the prior works [8, 10], we construct prompts using predefined templates such as "*<text> Describe this text in one word:*" for text and "*<image> Describe this image in one word:*" for image. These prompts are used to process the image-text pairs, then we extract the corresponding text embeddings $\{z_i^{\text{txt}}\}_{i=1}^N$ and image embeddings $\{z_i^{\text{img}}\}_{i=1}^N$.

**Contrastive Loss.** Once obtaining normalized embeddings $\{z_i^{\text{txt}}\}_{i=1}^N$ and $\{z_i^{\text{img}}\}_{i=1}^N$, we reformulate the standard autoregressive training paradigm of LLMs into a discriminative framework via a symmetric InfoNCE-style contrastive loss:

$$\mathcal{L}_{\text{contrast}} = \frac{1}{2N} \sum_{i=1}^N \left[ -\log \frac{\exp(z_i^{\text{img}} \cdot z_i^{\text{txt}}/\tau)}{\sum_{j=1}^N \exp(z_i^{\text{img}} \cdot z_j^{\text{txt}}/\tau)} - \log \frac{\exp(z_i^{\text{txt}} \cdot z_i^{\text{img}}/\tau)}{\sum_{j=1}^N \exp(z_i^{\text{txt}} \cdot z_j^{\text{img}}/\tau)} \right] \tag{1}$$

where $\tau$ is a temperature hyperparameter. Prior works [18, 19] established contrastive learning inherently employs a coarse-grained alignment strategy that uniformly pushes away all negative samples in the embedding space, with limited consideration of fine-grained semantic similarity between these samples. This uniform treatment struggles to establish nuanced discriminative boundaries, particularly for semantically similar negatives.

**Computing the Pairwise Alignment Score.** To probe the implicit modality alignment within an off-the-shelf MLLM, we measure the alignment score between an image-text pair $(x_i^{\text{img}}, x_i^{\text{txt}})$ based on the MLLM's output logits for "Yes" ($l_{ii}^{\text{Yes}}$) and "No" ($l_{ii}^{\text{No}}$) tokens in response to a relevance query (details in Appendix A.1). Then, we employ the softmax function for these two tokens to get the alignment score $\alpha_{ii}$, which represents the MLLM's confidence that the image $x_i^{\text{img}}$ and text $x_i^{\text{txt}}$ form a semantically matching pair.

**Measuring the Modality Gap.** The modality gap represents the misalignment between visual and textual feature distributions. Prior work [2] measured this using the average distance between mean embeddings: $\left\| \vec{\Delta}_{\text{gap}} \right\| = \|\mu_{\text{txt}} - \mu_{\text{img}}\|$, where $\mu_{\text{mod}} = \frac{1}{N} \sum_{i=1}^N z_i^{\text{mod}}$.

Expanding the squared distance, $\|\vec{\Delta}_{\text{gap}}\|^2 = \|\mu_{\text{txt}}\|^2 - 2\mu_{\text{txt}} \cdot \mu_{\text{img}} + \|\mu_{\text{img}}\|^2$, reveals that it measures the difference between the mean intra-modal similarity ($\frac{1}{N^2} \sum_{i=1}^N \sum_{j=1}^N z_i^{\text{mod}} \cdot z_j^{\text{mod}}$) and the mean cross-modal similarity ($\frac{1}{N^2} \sum_{i=1}^N \sum_{j=1}^N z_i^{\text{txt}} \cdot z_j^{\text{img}}$).

However, this mean-based comparison overlooks the full distributional characteristics of similarities. Furthermore, it requires explicit embeddings, making it less suitable to measure the modality gap for the logits-based model. To capture distributional discrepancy more effectively, we employ the 1-Wasserstein Distance (WD) [5] to compare similarity distributions. For two distributions $\mathbb{P}_A$ and $\mathbb{P}_B$, WD is defined as:

$$W(\mathbb{P}_A, \mathbb{P}_B) = \inf_{\gamma \in \Pi(\mathbb{P}_A, \mathbb{P}_B)} \mathbb{E}_{(s_a, s_b) \sim \gamma}[\|s_a - s_b\|] \tag{2}$$

where $s_a, s_b$ are samples from the distributions, and $\Pi(\cdot, \cdot)$ is the set of joint distributions with the given marginals.

We measure the modality gap using WD on the Winoground-style [14] fine-grained dataset, where each test instance contains two images ($x_0^{\text{img}}, x_1^{\text{img}}$) and two captions ($x_0^{\text{txt}}, x_1^{\text{txt}}$), forming two matching pairs ($x_0^{\text{img}}, x_0^{\text{txt}}$) and ($x_1^{\text{img}}, x_1^{\text{txt}}$). We denote the sets of all $x_0^{\text{txt}}$, $x_1^{\text{txt}}$, $x_0^{\text{img}}$, and $x_1^{\text{img}}$ as $T_0$, $T_1$, $I_0$, and $I_1$ respectively. We consider it from two perspectives: the distributional gap $W(\mathbb{P}_{T_0 I_0}, \mathbb{P}_{T_0 T_0})$,

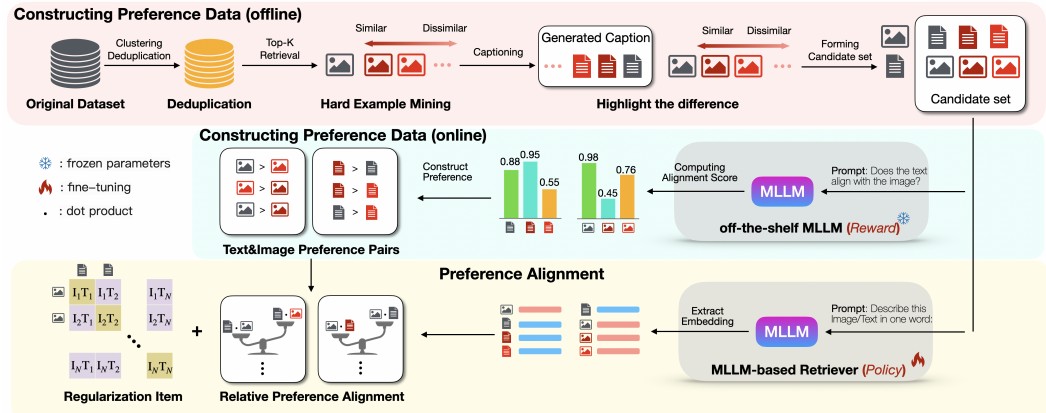

Figure 2: **The Training Schema of the Proposed MAPLE.** We first prepare the candidate set for each anchor sample through a series of dataset processing operations. In the training stage, we leverage an off-the-shelf MLLM as a reward model to dynamically calculate the alignment scores and subsequently construct the preference data. We extract the embeddings from the policy model (MLLM-based retriever) and align them with the preference data through the RPA loss. This schema primarily illustrates the pairwise training paradigm.

which quantifies the alignment between intra-modal $\mathbb{P}_{T_0 T_0}$ and cross-modal $\mathbb{P}_{T_0 I_0}$ similarity distributions (where a lower value indicates better alignment but risks representation collapse [20] when approaching zero), and the discriminative gap $W\left(\mathbb{P}_{T_0 I_0}, \mathbb{P}_{T_0 I_1}\right)$, which assesses the model's ability to distinguish between matching and non-matching pairs (where a higher value indicates better discriminative ability). Figure 3 in the Appendix illustrates the modality gap comparison between CLIP and Qwen2-VL based on this unified metric.

Finally, distributional gap $W_{\text{dist-gap}}$ and discriminative gap $W_{\text{disc-gap}}$ are computed as the mean values across all respective gap measurements in the dataset. We integrate both perspectives on the modality gap into our proposed metric ($\Delta_{\text{gap}}$): $\Delta_{\text{gap}} = W_{\text{dist-gap}}/W_{\text{disc-gap}}$. A lower $\Delta_{\text{gap}}$ reflects a superior model, characterized by both a small distributional gap and a large discriminative gap.

## 3 Method

We propose MAPLE (Modality-Aligned Preference Learning for Embeddings), a novel framework that guides cross-modal representations with MLLM priors via preference alignment. As illustrated in Figure 2, MAPLE consists of two key components: (1) ***Preference Data Construction***: An offline stage retrieves hard negative samples, followed by an online process where an off-the-shelf MLLM dynamically computes text-image alignment scores during training to establish preference data; and (2) ***Preference Alignment***: Derived from the DPO loss, we introduce a novel *Relative Preference Alignment* (RPA) loss that explicitly enhances the model's nuanced discriminative capability by contrasting preferred and dispreferred data samples.

**MLLM-based Retriever Architecture.** We initialize our model from a pretrained MLLM backbone to inherit its multi-modal alignment capability. Refer to the work [21], to convert the autoregressive MLLM into a discriminative retrieval paradigm, we make two small modifications: replacing the causal attention mask with bidirectional attention and adding mean-pooling over final hidden states to aggregate sufficient features for retrieval.

### 3.1 Preference Data Construction

The preference data construction involves two stages: an offline preparation stage and an online scoring and structuring stage.

**Offline Stage: Candidate Generation.** The pipeline begins with an offline stage to prepare candidate sets. We first extract DINOv2 [22] embeddings from the image dataset and apply Semantic Deduplication (SemDeup) [23]—a clustering-based method—to filter near-duplicate samples. For each deduplicated image $x_i^{\text{img}}$ in a batch, we retrieve its top-$K$ nearest neighbors $\{\hat{x}_j^{\text{img}}\}_{j=1}^K$ from the gallery based on cosine similarity, forming an image candidate set $\mathcal{C}_i^{\text{img}} = \{x_i^{\text{img}}\} \cup \{\hat{x}_j^{\text{img}}\}_{j=1}^K$. To enrich caption diversity and create challenging negatives, we leverage the multi-image reasoning capability of an off-the-shelf MLLM. We prompt the MLLM with the image set $\mathcal{C}_i^{\text{img}}$ to generate discriminative captions that explicitly highlight inter-image differences. This process yields a corresponding text candidate set $\mathcal{C}_i^{\text{txt}} = \{x_i^{\text{txt}}\} \cup \{\hat{x}_j^{\text{txt}}\}_{j=1}^K$. Candidate generation details are provided in Appendix B.1.

**Online Stage: Scoring and Structuring Preferences.** During the online training phase, we use the off-the-shelf MLLM to dynamically compute fine-grained alignment scores between anchors and candidates. For each anchor image $x_i^{\text{img}}$, we compute alignment scores with all text candidates: $\boldsymbol{\alpha}_i^{\text{img2txt}} = \{\text{align}(x_i^{\text{img}}, x) \mid x \in \mathcal{C}_i^{\text{txt}}\}$. Symmetrically, for each anchor text $x_i^{\text{txt}}$, we compute scores with all image candidates: $\boldsymbol{\alpha}_i^{\text{txt2img}} = \{\text{align}(x_i^{\text{txt}}, x) \mid x \in \mathcal{C}_i^{\text{img}}\}$. Each score vector $\boldsymbol{\alpha}_i$ (either $\boldsymbol{\alpha}_i^{\text{img2txt}}$ or $\boldsymbol{\alpha}_i^{\text{txt2img}}$) represents the MLLM's preference for candidates in $\mathcal{C}_i$ relative to the anchor $x_i$. We sort the candidates in descending order based on these scores, obtaining ranked indices $\{r_k\}_{k=0}^K$ such that the corresponding scores satisfy $\alpha_{i,r_0} \geq \alpha_{i,r_1} \geq \cdots \geq \alpha_{i,r_K}$. Based on this ranking, we structure the preference data in two ways for the subsequent alignment loss:

- **Pairwise Preferences:** We construct a set of preference pairs $\mathcal{P}_i = \{(x_{i,r_a}, x_{i,r_b}) \mid 0 \leq a < b \leq K\}$, where $x_{i,r_a}$ is the $r_a$-th ranked candidate and $x_{i,r_b}$ is the $r_b$-th ranked candidate from the corresponding set $\mathcal{C}_i$. Each pair $(x_{i,r_a}, x_{i,r_b})$ indicates that $x_{i,r_a}$ is preferred over $x_{i,r_b}$ according to the MLLM's alignment score with the anchor $x_i$.

- **Listwise Preferences:** Instead of breaking the ranking into independent pairs, we leverage the structure of the entire ranked list $(x_{i,r_0}, x_{i,r_1}, \ldots, x_{i,r_K})$. This approach considers preferences within all possible suffixes of the list. Specifically, for each starting rank $k$ (from 0 to $K-1$), the item $x_{i,r_k}$ is treated as the preferred item relative to the set of all subsequent items $\{x_{i,r_j}\}_{j=k+1}^K$ in the suffix $(x_{i,r_k}, \ldots, x_{i,r_K})$. This captures the relative ordering across the whole list more directly than pairwise comparisons.

Through these offline and online stages, for a batch of $N$ anchor image-text pairs $\{(x_i^{\text{img}}, x_i^{\text{txt}})\}_{i=1}^N$, we construct the corresponding sets of pairwise preferences and listwise preferences.

## 3.2 Preference Alignment

In contrast to the contrastive loss's coarse-grained alignment approach, our goal is to establish nuanced discriminative boundaries leveraging fine-grained preference data. Drawing inspiration from Direct Preference Optimization (DPO) [11], which effectively fine-tunes LLMs to align with human preferences, we similarly aim to fine-tune our MLLM-based retrieval model to align with the sophisticated preference signals from off-the-shelf MLLMs.

**DPO Loss.** Traditional DPO Loss relies on pairwise comparisons between preferred ($y_w$) and dispreferred ($y_l$) outputs for a given input $x$ to align policy models ($\pi_\theta$) with human preferences, often using a reference model ($\pi_w$). The DPO training objective is constructed as a maximum likelihood loss:

$$\mathcal{L}_{\text{DPO}}(\pi_\theta; \pi_w) = -\mathbb{E}_{(x,y_w,y_l)\sim\mathcal{D}}\left[\log \sigma\left(\beta \log \frac{\pi_\theta(y_w \mid x)}{\pi_w(y_w \mid x)} - \beta \log \frac{\pi_\theta(y_l \mid x)}{\pi_w(y_l \mid x)}\right)\right] \quad (3)$$

Here, $\sigma$ denotes the sigmoid function, and $\beta$ is a hyperparameter scaling the log-probability difference. However, directly applying DPO to retrieval presents challenges: (1) The diverse captions generated in the offline stage (Section 3.1) create a combinatorial explosion of potential image-caption pairs, making it impractical to explicitly enumerate all preference pairs; (2) The standard DPO framework requires maintaining policy, reference, and potentially reward models, imposing substantial memory and computational burdens.

**Eliminating the Need for the Reference Model.** The memory-inefficiency issue can be alleviated by adopting a uniform prior $U$ for the reference model $\pi_w$, similar to CPO [24]. In this case, the reference model terms $\pi_w(y_w \mid x)$ and $\pi_w(y_l \mid x)$ cancel out in the log-ratio difference (up to a constant), eliminating the need for costly computations and storage associated with $\pi_w$. The simplified objective becomes:

$$\mathcal{L}_{\text{DPO-simplified}}(\pi_\theta) = -\mathbb{E}_{(x,y_w,y_l)\sim\mathcal{D}}\left[\log\sigma\left(\beta\log\pi_\theta(y_w \mid x) - \beta\log\pi_\theta(y_l \mid x)\right)\right] \tag{4}$$

**Relative Preference Alignment (RPA) Loss.** We adapt the simplified DPO objective for embedding models by replacing the log probabilities $\log\pi_\theta(y \mid x)$ with scaled similarity scores $\beta(z^{\text{anchor}}\cdot z^{\text{candidate}})$ between anchor and candidate embeddings produced by our MLLM-based retrieval model. This approach, which we term **Relative Preference Alignment (RPA)**, incorporates the MLLM-derived preference structures (pairwise or listwise). We explore two primary strategies for implementing RPA: pairwise and listwise optimization.

The **Pairwise RPA** loss optimizes preferences using the pairwise data $\mathcal{P}_i$. For a given text anchor $x_i^{\text{txt}}$ and a preference pair $(x_{i,r_k}^{\text{img}}, x_{i,r_l}^{\text{img}}) \in \mathcal{P}_i^{\text{txt2img}}$ (where $k < l$), it aims to ensure the similarity score of the preferred image is higher than the dispreferred one. Let $s_{ik}^{\text{txt2img}} = \beta(z_i^{\text{txt}} \cdot z_{i,r_k}^{\text{img}})$ denote the scaled similarity score. The pairwise RPA loss for text-to-image alignment (txt2img) is weighted by the difference between the MLLM's alignment scores for the pair, giving more importance to pairs with larger preference margins:

$$\mathcal{L}_{\text{RPA-Pairwise}}^{\text{txt2img}} = -\frac{1}{N}\sum_{i=1}^{N}\sum_{0\leq k<l\leq K}(\alpha_{i,r_k}^{\text{txt2img}} - \alpha_{i,r_l}^{\text{txt2img}})\log\sigma(s_{ik}^{\text{txt2img}} - s_{il}^{\text{txt2img}}) \tag{5}$$

Symmetrically, we define the image-to-text loss $\mathcal{L}_{\text{RPA-Pairwise}}^{\text{img2txt}}$ using scores $s_{ik}^{\text{img2txt}} = \beta(z_i^{\text{img}} \cdot z_{i,r_k}^{\text{txt}})$ and pairs from $\mathcal{P}_i^{\text{img2txt}}$. The total pairwise RPA loss is:

$$\mathcal{L}_{\text{RPA-Pairwise}} = \frac{1}{2}(\mathcal{L}_{\text{RPA-Pairwise}}^{\text{txt2img}} + \mathcal{L}_{\text{RPA-Pairwise}}^{\text{img2txt}}) \tag{6}$$

Alternatively, the **Listwise RPA** approach directly optimizes the model's ability to align with the MLLM's ranking using the listwise preference data. Inspired by PRO [25], this loss encourages the model to assign the highest similarity score to the top-ranked item within each suffix of the MLLM's ranked list. For a text anchor $x_i^{\text{txt}}$ and its ranked image candidates $(x_{i,r_0}^{\text{img}}, \ldots, x_{i,r_K}^{\text{img}})$, the loss iterates through each possible top element $x_{i,r_k}^{\text{img}}$ (for $k$ from 0 to $K-1$) and maximizes the log-probability of this element being ranked highest among the suffix $\{x_{i,r_j}^{\text{img}}\}_{j=k}^{K}$, using a softmax over the model's similarity scores $s_{ij}^{\text{txt2img}}$. To incorporate the fine-grained preference strength from the MLLM, each term in the sum (corresponding to a specific suffix starting at $k$) is weighted by the average preference margin assigned by the MLLM to $x_{i,r_k}^{\text{img}}$ over the subsequent items in that suffix:

$$\mathcal{L}_{\text{RPA-Listwise}}^{\text{txt2img}} = -\frac{1}{N}\sum_{i=1}^{N}\sum_{k=0}^{K-1}w_{ik}^{\text{txt2img}}\log\frac{\exp(s_{ik}^{\text{txt2img}})}{\sum_{j=k}^{K}\exp(s_{ij}^{\text{txt2img}})} \tag{7}$$

where the weight $w_{ik}^{\text{txt2img}} = \frac{1}{K-k}\sum_{l=k+1}^{K}(\alpha_{i,r_k}^{\text{txt2img}} - \alpha_{i,r_l}^{\text{txt2img}})$ is the average MLLM alignment score difference between candidate $k$ and all less preferred candidates (ranks $k+1$ to $K$) in the list (defined as 0 if $k = K$). This weighting gives more importance to correctly ranking the top element in suffixes where the MLLM preference is strong and clear. The corresponding image-to-text loss, $\mathcal{L}_{\text{RPA-Listwise}}^{\text{img2txt}}$, is defined symmetrically using $s_{ik}^{\text{img2txt}}$ and weights $w_{ik}^{\text{img2txt}}$ derived from $\boldsymbol{\alpha}_i^{\text{img2txt}}$. The total listwise RPA loss is:

$$\mathcal{L}_{\text{RPA-Listwise}} = \frac{1}{2}(\mathcal{L}_{\text{RPA-Listwise}}^{\text{txt2img}} + \mathcal{L}_{\text{RPA-Listwise}}^{\text{img2txt}}) \tag{8}$$

**Regularized Relative Preference Alignment.** To prevent excessive alignment to the MLLM preferences, which might lead to feature collapse, we introduce a regularization term $\mathcal{L}_{\text{contrast}}$. This is typically a standard contrastive loss computed on the original anchor pairs $(x_i^{\text{img}}, x_i^{\text{txt}})$ within the batch. The final training objective combines the chosen RPA loss (either pairwise or listwise) with this regularization:

$$\mathcal{L} = \lambda\mathcal{L}_{\text{RPA}} + (1 - \lambda)\mathcal{L}_{\text{contrast}} \tag{9}$$

where $\mathcal{L}_{\text{RPA}}$ represents the chosen RPA loss component (either $\mathcal{L}_{\text{RPA-Listwise}}$ or $\mathcal{L}_{\text{RPA-Pairwise}}$), and $\lambda$ serves as a balancing hyperparameter controlling the strength of the preference alignment relative to the regularization term.

# 4 Experiments & Results

In this section, we evaluate MAPLE across multiple standard benchmarks to assess its effectiveness in cross-modal retrieval tasks. We compare MAPLE against strong baselines and analyze its performance through ablation studies.

## 4.1 Experimental Setup

We train MAPLE on a curated subset of the OpenImage dataset [26] (details in Appendix B.1). We evaluate its performance on standard general (MS-COCO [12], Flickr30K [13]) and fine-grained (Winoground [14], NaturalBench [15], MMVP [16], BiVLC [17]) retrieval benchmarks using standard metrics (e.g., Image/Text Recall@1, Image/Text scores). About the more fine-grained evaluation details, please refer to Appendix B.3. MAPLE is compared against strong CLIP-based [1, 27–30] and MLLM-based [8, 10] retrieval models. We use LoRA for fine-tuning MAPLE MLLM-based embedding models. Comprehensive implementation details are provided in Appendix B.2.

## 4.2 Main Results

**Performance on Retrieval Benchmarks.** As shown in Table 1, our MAPLE(*Qwen2-VL-7B*) approach consistently outperforms both CLIP-based and MLLM-based models across multiple benchmarks. Under the similar-scale parameters settings, MAPLE(*Qwen2-VL-2B*) also shows a competitive advantage. The improvement is even more pronounced on fine-grained retrieval tasks, where MAPLE demonstrates substantial gains on Winoground and NaturalBench, significantly outperforming previous methods. These results validate the effectiveness of our preference-guided alignment approach in capturing nuanced cross-modal relationships.

## 4.3 Ablation Studies

To rigorously evaluate our proposed method MAPLE and dissect the contributions of its key components, we conduct a series of ablation studies. Our final proposed loss function combines a standard contrastive loss $\mathcal{L}_{\text{contrast}}$ with our novel RPA loss $\mathcal{L}_{\text{RPA}}$. The standard contrastive loss $\mathcal{L}_{\text{contrast}}$ is computed using image-text anchor pairs gathered across all devices to maintain the model's general cross-modal alignment capabilities, serving as a regularization term to prevent excessive alignment. Meanwhile, $\mathcal{L}_{\text{RPA}}$ operates specifically on preference data derived from retrieved examples, aiming to refine the model's understanding of fine-grained distinctions. We investigate the impact of using these components individually and in combination.

**Relative Preference Alignment.** We analyze the impact of different loss components, referencing Table 2. The baseline uses only the standard contrastive loss ($\mathcal{L}_{\text{contrast}}$).

Compared with the baseline, using only a standard contrastive loss on preference examples ($\mathcal{L}_{\text{contrast-pref}}$) generally degrades general retrieval performance and shows moderate improvements on the NaturalBench dataset, with only slight improvements on the Winoground Image task. In contrast, applying our RPA losses ($\mathcal{L}_{\text{RPA-Pairwise}}$, $\mathcal{L}_{\text{RPA-Listwise}}$) directly to preference data, despite reducing general performance when used alone, achieves substantially better fine-grained results than

Table 1: **Performance Comparison on General and Fine-grained Retrieval Tasks.** Best results are in **bold**, second best are underlined. VladVA haven't released the model weights, so "−" represents unavailable results.

| Model | General Retrieval (R@1) | | | | Fine-grained Retrieval | | | |
| | COCO | | Flickr30k | | Winoground | | NaturalBench | |
| | Text | Image | Text | Image | Text | Image | Text | Image |
|---|---|---|---|---|---|---|---|---|
| **CLIP-based Models** | | | | | | | | |
| CLIP (*ViT-L*) [1] | 58.1 | 37.0 | 87.2 | 67.3 | 27.5 | 12.3 | 41.8 | 45.0 |
| OpenCLIP (*ViT-G/14*) [27] | 66.3 | 48.8 | 91.5 | 77.8 | 32.0 | 12.8 | 46.2 | 46.5 |
| SigLIP (*so/14*) [28] | 70.2 | 52.0 | 93.5 | 80.5 | 37.5 | 16.3 | 62.7 | 63.9 |
| SigLIPv2 (*g/16-2B*) [30] | 72.8 | 56.1 | **95.4** | 86.0 | 39.8 | 17.0 | 65.5 | 68.7 |
| EVA-CLIP (*8B*) [29] | 70.1 | 52.0 | 94.5 | 80.3 | 36.5 | 14.8 | 58.5 | 59.3 |
| EVA-CLIP (*18B*) [29] | 72.8 | 55.6 | 95.3 | 83.3 | 35.8 | 15.0 | 58.7 | 61.2 |
| **MLLM-based Models** | | | | | | | | |
| E5-V(*LLaVA-Next-8B*) [8] | 62.0 | 52.0 | 88.2 | 79.5 | 32.3 | 14.8 | 60.3 | 67.6 |
| VladVA(*Qwen2-VL-2B*) [10] | 71.9 | 52.5 | 93.7 | 80.4 | - | - | - | - |
| VladVA(*LLaVA-1.5-7B*) [10] | 72.9 | 59.0 | 94.3 | 83.3 | 40.5 | 17.5 | - | - |
| **MAPLE(*Qwen2-VL-2B*)** | 72.8 | 56.8 | 92.8 | 82.6 | 43.0 | 22.5 | 69.2 | 70.2 |
| **MAPLE(*Qwen2-VL-7B*)** | **75.5** | **60.3** | 94.3 | **86.1** | **56.0** | **32.7** | **76.1** | **76.8** |

Table 2: **Ablation Study on Loss Components.** Performance on General and Fine-grained Retrieval Tasks, compared to the baseline ($\mathcal{L}_{\text{contrast}}$). Differences are shown in parentheses with arrows (↑ for improvement, ↓ for decline).

| Method | General Retrieval | | Fine-grained Retrieval | | | |
| | COCO | | Winoground | | NaturalBench | |
| | Text | Image | Text | Image | Text | Image |
|---|---|---|---|---|---|---|
| **Baseline Contrastive Loss Only** | | | | | | |
| Baseline ($\mathcal{L}_{\text{contrast}}$) | 74.0 | 54.4 | 42.5 | 20.5 | 61.4 | 62.5 |
| **Training with Preference Data Only** | | | | | | |
| $\mathcal{L}_{\text{contrast-pref}}$ | 64.6 (↓-9.4) | 46.2 (↓-8.2) | 42.3 (↓-0.2) | 20.7 (↑+0.2) | 66.1 (↑+4.7) | 67.8 (↑+5.3) |
| $\mathcal{L}_{\text{RPA-Pairwise}}$ | 51.9 (↓-22.1) | 52.4 (↓-2.0) | 48.8 (↑+6.3) | 34.7 (↑+14.2) | 70.1 (↑+8.7) | 77.3 (↑+14.8) |
| $\mathcal{L}_{\text{RPA-Listwise}}$ | 57.1 (↓-16.9) | 55.4 (↑+1.0) | 48.0 (↑+5.5) | 36.5 (↑+16.0) | 71.1 (↑+9.7) | 78.1 (↑+15.6) |
| **Combining Preference Loss with Contrastive Regularizer** | | | | | | |
| $\mathcal{L}_{\text{contrast}} + \mathcal{L}_{\text{contrast-pref}}$ | 70.9 (↓-3.1) | 53.9 (↓-0.5) | 46.5 (↑+4.0) | 21.5 (↑+1.0) | 65.5 (↑+4.1) | 67.0 (↑+4.5) |
| $\mathcal{L}_{\text{contrast}} + \mathcal{L}_{\text{RPA-Pairwise}}$ | 71.2 (↓-2.8) | 57.7 (↑+3.3) | 49.8 (↑+7.3) | 26.8 (↑+6.3) | 68.6 (↑+7.2) | 71.4 (↑+8.9) |
| $\mathcal{L}_{\text{contrast}} + \mathcal{L}_{\text{RPA-Listwise}}$ | 71.9 (↓-2.1) | 58.6 (↑+4.2) | 51.0 (↑+8.5) | 28.2 (↑+7.7) | 69.2 (↑+7.8) | 71.2 (↑+8.7) |

$\mathcal{L}_{\text{contrast-pref}}$. This highlights the importance of introducing preference data to improve the model's ability to make nuanced distinctions.

We then combine preference-based losses with the standard contrastive loss ($\mathcal{L}_{\text{contrast}}$) as a regularizer. Through extensive experimentation, we identify the optimal $\lambda$ parameter for each combination that maximizes average performance on both general and fine-grained retrieval tasks. When combined with the $\mathcal{L}_{\text{contrast}}$ regularizer, this approach effectively mitigates performance degradation on general retrieval while maintaining strong fine-grained retrieval performance. Moreover, $\mathcal{L}_{\text{RPA}}$ consistently outperforms $\mathcal{L}_{\text{contrast-pref}}$. Additionally, we observe that $\mathcal{L}_{\text{RPA-Listwise}}$ consistently achieves better results than $\mathcal{L}_{\text{RPA-Pairwise}}$ across nearly all scenarios. We attribute this to the fact that listwise loss aligns preferences across the entire ranked list, making it more effective than pairwise comparisons.

**Impact of Expanded Negative Pool for Contrastive Loss.** Expanding batch size plays a critical role in contrastive learning. However, increasing batch size for MLLMs incurs substantial computational overhead. To address this challenge, we propose an efficient strategy that implicitly enlarges

Table 3: **Ablation Study on Using Expanded Negatives (Exp. Neg.) and $\mathcal{L}_{\text{RPA-Listwise}}$ ($\mathcal{L}_{\text{RPA}}$).** The baseline (✗/✗) uses neither component. ✓ indicates the component is used, ✗ indicates it is not. Best results are in **bold**.

| Components | | General Retrieval | Fine-grained Retrieval | | | |
|---|---|---|---|---|---|---|
| Exp. Neg. | $\mathcal{L}_{\text{RPA}}$ | COCO | Winoground | NaturalBench | BiVLC | MMVP |
| | | T / I | T / I | T / I | T / I | T / I |
| ✗ | ✗ | 74.0 / 54.4 | 42.5 / 20.5 | 61.4 / 62.5 | 86.1 / 60.5 | 33.3 / 20.0 |
| ✓ | ✗ | 75.3 / 57.3 | 49.3 / 24.8 | 70.7 / 70.2 | 89.2 / 66.6 | 39.3 / 34.1 |
| ✗ | ✓ | 71.9 / 58.6 | 51.0 / 28.2 | 69.2 / 71.2 | 88.3 / 73.4 | **46.7** / 37.0 |
| ✓ | ✓ | **75.5 / 60.3** | **56.0 / 32.7** | **76.1 / 76.8** | **89.4 / 75.5** | 45.9 / **43.7** |

Table 4: **Robustness to Reward Model Scale and Architecture.** The policy model is fixed to Qwen2-VL-7B. Best results within each model family are in **bold**. The '-' indicates the baseline without a reward model.

| Reward Model | General Retrieval | | | | Fine-grained Retrieval | | | |
|---|---|---|---|---|---|---|---|---|
| | COCO | | Flickr30k | | Winoground | | NaturalBench | |
| | Text | Image | Text | Image | Text | Image | Text | Image |
| - | 73.4 | 54.3 | 93.6 | 80.3 | 40.7 | 18.2 | 60.2 | 62.7 |
| **Qwen Family** | | | | | | | | |
| Qwen2-VL-2B | 74.1 | 59.1 | 93.0 | 84.1 | 53.5 | **31.0** | 70.4 | 72.3 |
| Qwen2-VL-7B | **75.8** | **60.2** | **94.2** | **85.3** | **55.0** | **31.0** | **74.5** | **75.2** |
| **InternVL3 Family** | | | | | | | | |
| InternVL3-1B | 73.9 | 58.9 | 92.8 | 83.7 | 48.5 | 26.0 | 69.3 | 72.6 |
| InternVL3-2B | **75.9** | 59.2 | **94.2** | **84.8** | 53.8 | 28.5 | 72.8 | 74.3 |
| InternVL3-8B | 75.6 | **59.7** | 93.8 | **84.8** | **54.0** | **31.5** | **74.3** | **74.8** |
| **Additional Architectures** | | | | | | | | |
| SAIL-VL-1.6-8B | 76.1 | 59.9 | 94.3 | 85.6 | 54.8 | 29.8 | 75.4 | 74.3 |
| InternVL2.5-8B | 75.2 | 59.7 | 94.3 | 85.3 | 54.8 | 30.5 | 74.6 | 74.8 |
| InternVL2.5-8B-MPO | 75.5 | 59.5 | 93.8 | 85.1 | 53.5 | 30.8 | 74.1 | 74.2 |

the effective batch size through the incorporation of readily available hard negatives, eliminating the need for additional computational resources. The detailed implementation of this strategy is provided in Appendix C.1. Empirical results in Table 3 demonstrate that our expanded negative pool strategy yields consistent performance improvements across both general and fine-grained retrieval tasks.

**Robustness to Reward Model Choice.** To assess the generalizability and robustness of MAPLE, we conduct a comprehensive ablation study on the choice of the reward model. We fix the policy model to Qwen2-VL-7B and evaluate its performance when guided by reward signals from a diverse set of MLLMs, varying in both architectural family and scale. For quick comparison, all experiments in this ablation were trained for 4 epochs. As shown in Table 4, our analysis reveals that MAPLE is robust to the scale of the reward model. While larger reward models generally yield stronger results, even small models like Qwen2-VL-2B and InternVL3-1B lead to consistent and significant performance gains over the baseline. This result underscores the framework's resource efficiency, as it demonstrates that substantial performance gains can be achieved by leveraging alignment signals from even small-scale reward models.

**Agnostic to Reward Model Architecture.** Furthermore, Table 4 shows that MAPLE is agnostic to the reward model's architecture. Using reward models from different architectural families (e.g., InternVL3 [31], InternVL2.5 [32], InternVL2.5-MPO [33], and SAIL-VL [34]) to guide the Qwen2-VL-7B policy model still leads to clear improvements over the baseline. To further demonstrate this, we conduct a systematic cross-architecture evaluation in Table 5, where both the policy and reward models are varied. The results confirm that even mismatched reward-policy pairs are highly effective,

Table 5: **Cross-Architecture Robustness Analysis.** Performance of MAPLE with varying policy and reward model architectures. The '-' indicates the baseline without a reward model. The best results are in **bold**, the second best are underlined.

| Policy Model | Reward Model | COCO | Flickr30k | Winoground | NaturalBench |
|---|---|---|---|---|---|
| | | T / I | T / I | T / I | T / I |
| Qwen2-VL-7B | - | 73.4 / 54.3 | 93.6 / 80.3 | 40.7 / 18.2 | 60.2 / 62.7 |
| | Qwen2-VL-7B | 75.8 / 60.2 | 94.2 / 85.3 | **55.0** / 31.0 | 74.5 / 75.2 |
| | InternVL3-8B | 75.6 / 59.7 | 93.8 / 84.8 | 54.0 / **31.5** | 74.3 / 74.8 |
| InternVL3-8B | Qwen2-VL-7B | 76.7 / 61.1 | 95.4 / 86.8 | 53.5 / **31.5** | **78.4** / 77.7 |
| | InternVL3-8B | **76.9** / **61.6** | **95.9** / **87.5** | 53.5 / 30.5 | **78.4** / **79.1** |

Table 6: **Comparison of Modality Gap Metrics Across Different Models.** Values of $W_{\text{dist-gap}}$ and $W_{\text{disc-gap}}$ are scaled by $10^2$. ↑ indicates higher values are better, ↓ indicates lower values are better. About the $\Delta_{gap}$ metric, the best results are in **bold**, the second best are underlined.

| Model | MMVP | | | Winoground | | |
|---|---|---|---|---|---|---|
| | $W_{\text{dist-gap}}$ (↓) | $W_{\text{disc-gap}}$ (↑) | $\Delta_{gap}$ (↓) | $W_{\text{dist-gap}}$ (↓) | $W_{\text{disc-gap}}$ (↑) | $\Delta_{gap}$ (↓) |
| OpenCLIP (*ViT-H/14*) [27] | 23.64 | 0.50 | 47.28 | 18.05 | 0.50 | 36.1 |
| Qwen2-VL-7B [6] | 6.89 | 6.34 | **1.09** | 7.63 | 2.13 | **3.58** |
| **MAPLE (w/o $\mathcal{L}_{\text{RPA-Listwise}}$)** | 6.32 | 0.33 | 19.15 | 4.40 | 0.44 | 10.0 |
| **MAPLE (w $\mathcal{L}_{\text{RPA-Listwise}}$)** | 5.32 | 0.94 | 5.66 | 3.47 | 0.48 | 7.22 |

which underscores the framework's flexibility and high degree of interoperability. This suggests that MAPLE learns a truly transferable, architecture-agnostic alignment signal, rather than simply overfitting to the intrinsic biases of a specific reward model architecture.

**Measuring the Modality Gap.** We measure the gap on the MMVP and Winoground datasets, where for the Winoground dataset we randomly sample 100 instances (out of 400 total instances) due to computational constraints in calculating pairwise alignment scores with MLLM. Table 6 presents modality gap metrics across various models. The off-the-shelf MLLM (Qwen2-VL-7B) demonstrates strong discriminative capability (having the largest $W_{\text{disc-gap}}$), while maintaining a low distributional gap. MAPLE with RPA loss substantially reduces the distributional gap while improving discriminative power. The evolution of this modality gap throughout the training process is visualized in Figure 6 in Appendix C.

## 5 Conclusion

In this work, using a unified metric based on the Wasserstein distance, we reveal that Multimodal Large Language Models (MLLMs) possess strong inherent modality alignment capabilities. Motivated by this finding, we explore leveraging off-the-shelf MLLM to mitigate the modality gap in cross-modal retrieval. We introduce MAPLE, a novel training framework designed to transfer the modality alignment priors of MLLM into corss-modal representations. MAPLE first constructs preference data via an automatic curation pipeline and subsequently employs a novel Relative Preference Alignment (RPA) loss tailored for MLLM-based retrieval model. Extensive experiments across various benchmarks demonstrate that MAPLE brings significant improvements on fine-grained tasks. Ablation studies validate the effectiveness of each component within MAPLE.

**Impact and Limitations:** Our work may contribute to exploring new directions for guiding cross-modal representations with alignment knowledge from MLLMs and potentially offers insights for future research in multimodal representation learning and preference-guided optimization. However, we acknowledge several limitations. Known limitations of this work include: 1) The cross-modal representations may be affected by the inherent biases present in the MLLM; 2) our approach requires further validation on more complex tasks such as composed retrieval. We plan to address these limitations in future work.

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

# A  Preliminaries

## A.1  Detailed Pairwise Alignment Score Computation.

Recent works [35, 36] prompt LLMs to make binary relevance judgments and construct candidate rankings. Inspired by these works, we leverage MLLMs to generate fine-grained alignment scores. For each image-text pair, we construct a specific relevance prompt "*<image> Does the image align with the text <text>? Answer Yes or No*". This prompt, containing both the image and the text, is fed into the MLLM. The MLLM then processes the input and generates token logits, including those for the target tokens "Yes" and "No".

We then calculate the pairwise alignment score, denoted as $\alpha_{ii}$, by applying the softmax function specifically to the logits of the "Yes" and "No" tokens. This normalization yields the probability of the "Yes" token:

$$\alpha_{ii} = \text{align}(x_i^{\text{img}}, x_i^{\text{txt}}) = \frac{\exp(l_{ii}^{\text{Yes}})}{\exp(l_{ii}^{\text{Yes}}) + \exp(l_{ii}^{\text{No}})} \tag{10}$$

Here, $l_{ii}^{\text{Yes}}$ and $l_{ii}^{\text{No}}$ represent the output logits produced by the MLLM for the "Yes" and "No" tokens, respectively, in response to the relevance prompt for the specific pair $(x_i^{\text{img}}, x_i^{\text{txt}})$. The resulting score $\alpha_{ii}$ serves as a quantitative measure of the MLLM's semantic confidence that the given image and text constitute a matching multimodal pair.

# B  Detailed Experiment Settings

## B.1  Detailed Training Data Curation

Our primary training data originates from the large-scale OpenImage dataset [26]. To curate a high-quality dataset suitable for fine-grained learning, we employed the following procedure:

1. **Clustering and Deduplication:** We first applied clustering and deduplication to the Human-verified OpenImage data to reduce redundancy and group similar images. Following the SemDeDup method [23], we set the number of clusters to 50,000 and used an epsilon value of 0.07 to filter out near-duplicate images.

2. **Hard Negative Mining:** For each anchor image in the deduplicated set, we identified hard negative images by retrieving the top-3 most similar images based on their DINOv2 [22] embeddings.

3. **Stratified Sampling:** We performed stratified sampling based on the clustering results to balance data diversity with training resource constraints. The dataset was divided into 8 partitions, and we sampled only one partition (approximately 700K instances) for our training set.

4. **Caption Generation:** For each anchor image and its selected top-3 hard negative images, we employed Qwen2.5-VL-72B [7] to generate corresponding captions for paired images (constructing 6 distinct image pairs from each anchor image and its top-3 hard negative images). To enhance descriptive diversity, we configured the generation temperature to 0.7 and applied the generation process three times, resulting in rich and varied textual representations. As demonstrated in Figure 4, this method enabled us to generate detailed comparative descriptions that effectively capture the visual distinctions between images.

The final training instances thus consist of an anchor image, its associated positive caption(s), and the captions corresponding to its identified hard negative images. While alternative data curation strategies might further enhance model performance, our work primarily focuses on preference alignment optimization; thus, exploring advanced data strategies is left for future work.

## B.2  Detailed Implementation Settings

MAPLE consists of two models: a reward model (Qwen2-VL-7B) and a policy model (initialized with Qwen2-VL-2B or Qwen2-VL-7B). We fine-tune the policy model using Low-Rank Adaptation

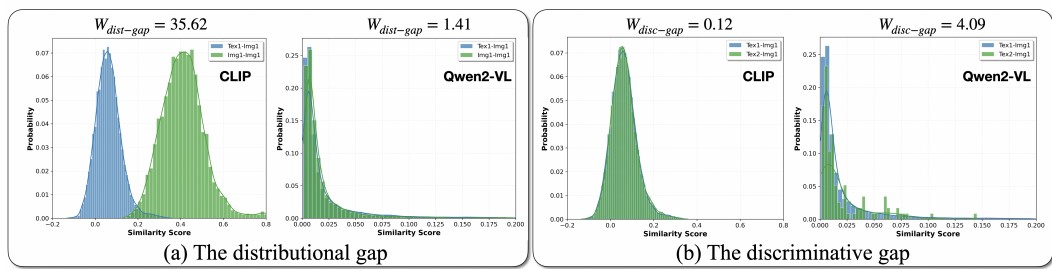

Figure 3: **The Modality Gap Comparison Between CLIP and Qwen2-VL.** The gap is computed on the MMVP dataset. For the CLIP model, we use cosine similarity to construct the similarity distribution. For the Qwen2-VL model, we use the alignment score to construct the similarity distribution. $W_{\text{dist-gap}}$ indicates lower values are better, $W_{\text{disc-gap}}$ indicates higher values are better.

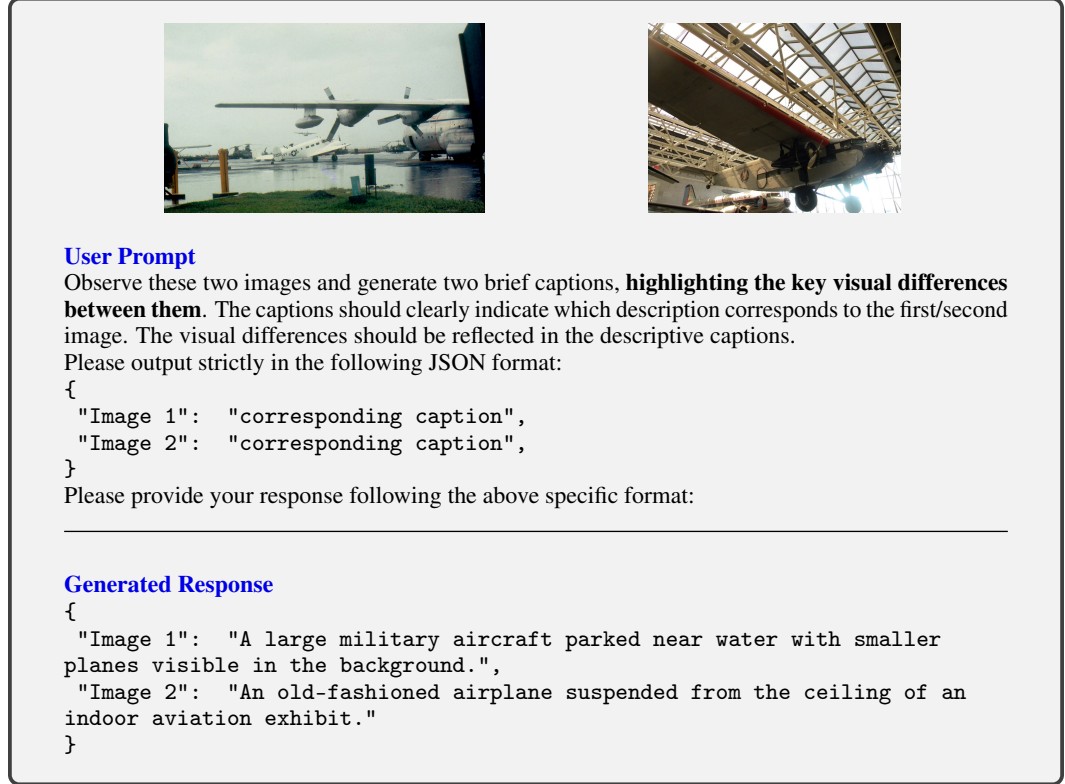

Figure 4: **Example of Prompt and Response for Generating Comparative Descriptions.** An example illustrating a prompt for generating comparative descriptions for a pair of images and the corresponding JSON response generated by the Qwen2.5-VL-72B.

(LoRA) [37], applying it only to the LLM component's attention and projection layers, while keeping the vision encoder and connector frozen. We employ a differential learning rate strategy, with the base learning rate applied to the LoRA parameters and amplifying the base learning rate by 100 times for the newly added parameters $\tau$ and $\beta$. Key model-related hyperparameters are summarized in Table 7.

The contrastive loss $\mathcal{L}_{\text{contrast}}$ temperature $\tau$ and the $\mathcal{L}_{\text{RPA}}$ coefficient $\beta$ are treated as learnable parameters, initialized at $0.07$ and $1/0.07$ respectively.

**Infrastructure and Efficiency.** Training was conducted on a cluster of 32 NVIDIA A100 GPUs (80GB memory each). To optimize computational efficiency and memory usage, we employed

Table 7: **Key Hyperparameters for MAPLE Training.**

| Parameter | Value |
|---|---|
| Policy Model Base | Qwen2-VL-2B / Qwen2-VL-7B |
| LoRA Rank ($r$) | 32 |
| LoRA Alpha ($\alpha$) | 32 |
| Optimizer | AdamW |
| Base Learning Rate (LoRA) | 5e-4 |
| LR Schedule | Linear Warmup + Cosine Decay |
| Warmup Ratio | 0.025 (of total steps) |
| Batch Size (per GPU) | 96 (for 2B model) / 48 (for 7B model) |
| Total Epochs | 8 |
| Max Image Resolution | 384x384 |
| Initial $\tau$ (learnable) | 0.07 |
| Initial $\beta$ (learnable) | $1/0.07 \approx 14.29$ |

bfloat16 mixed-precision training, gradient checkpointing for the LLM component, and Flash Attention [38]. The maximum image resolution was constrained to 384×384 during training to manage memory and allow for larger batch sizes. The whole training takes about 32 hours in this setting.

### B.3 Detailed Fine-grained Evaluation Dataset and Metrics

To comprehensively evaluate fine-grained capabilities, we employ a diverse set of benchmarks that assess compositional reasoning and subtle visual distinctions:

- **Winoground [14]**: Tests compositional reasoning abilities through 400 carefully designed instances that require understanding nuanced relationships between objects in images and their textual descriptions.

- **NaturalBench [15]**: An expanded benchmark containing 1,200 instances that builds upon Winoground's principles with greater diversity.

- **MMVP [16]**: Comprises 135 instances distributed across 9 distinct visual reasoning categories: Orientation and Direction, Presence of Specific Features, State and Condition, Quantity and Count, Positional and Relational Context, Color and Appearance, Structural and Physical Characteristics, Text, and Viewpoint and Perspective.

- **BiVLC [17]**: A large-scale benchmark with 2,933 instances, each categorized into one of three transformation types (Replace, Swap, or Add) that test different aspects of text-image alignment.

**Evaluation protocol.** Each test instance in the above datasets contains two images $(x_0^{\text{img}}, x_1^{\text{img}})$ and two captions $(x_0^{\text{txt}}, x_1^{\text{txt}})$, forming two correct pairs $(x_0^{\text{img}}, x_0^{\text{txt}})$ and $(x_1^{\text{img}}, x_1^{\text{txt}})$. The goal is to correctly associate images with their corresponding captions. The Image score measures whether the model correctly identifies the image for *both* captions: $\mathbf{1}[s(x_0^{\text{txt}}, x_0^{\text{img}}) > s(x_0^{\text{txt}}, x_1^{\text{img}}) \wedge s(x_1^{\text{txt}}, x_1^{\text{img}}) > s(x_1^{\text{txt}}, x_0^{\text{img}})]$, where $s(\cdot, \cdot)$ is the similarity function and $\mathbf{1}[\cdot]$ is the indicator function. Similarly, the Text score measures whether the model correctly identifies the caption for *both* images: $\mathbf{1}[s(x_0^{\text{img}}, x_0^{\text{txt}}) > s(x_0^{\text{img}}, x_1^{\text{txt}}) \wedge s(x_1^{\text{img}}, x_1^{\text{txt}}) > s(x_1^{\text{img}}, x_0^{\text{txt}})]$.

## C Supplementary Experimental Results

### C.1 Expanded Negative Pool for Contrastive Loss

Within a single device, instead of computing similarities solely among the $N$ anchor examples, we leverage the $K$ retrieved hard negatives associated with each anchor. This expands the effective pool of examples for contrastive comparison to $N(1 + K)$ per device. When extending this across multiple devices, we gather all anchor and hard negative examples globally. Since duplicates may arise in this aggregated set (e.g., the same hard negative might be retrieved for different anchors), we perform

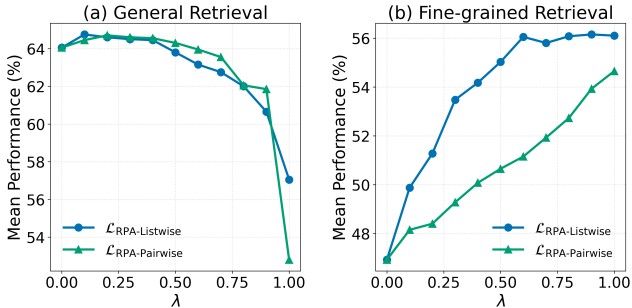
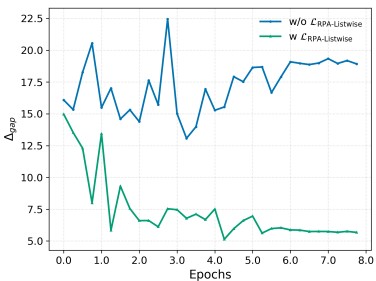

Figure 5: **Impact of Varying the Hyperparameter $\lambda$ on Retrieval Performance.** The mean of the y-axis represents the average performance across Text and Image retrieval tasks.

Figure 6: **Evolution of the Modality Gap Throughout the Training Process.** The gap is computed on the MMVP dataset.

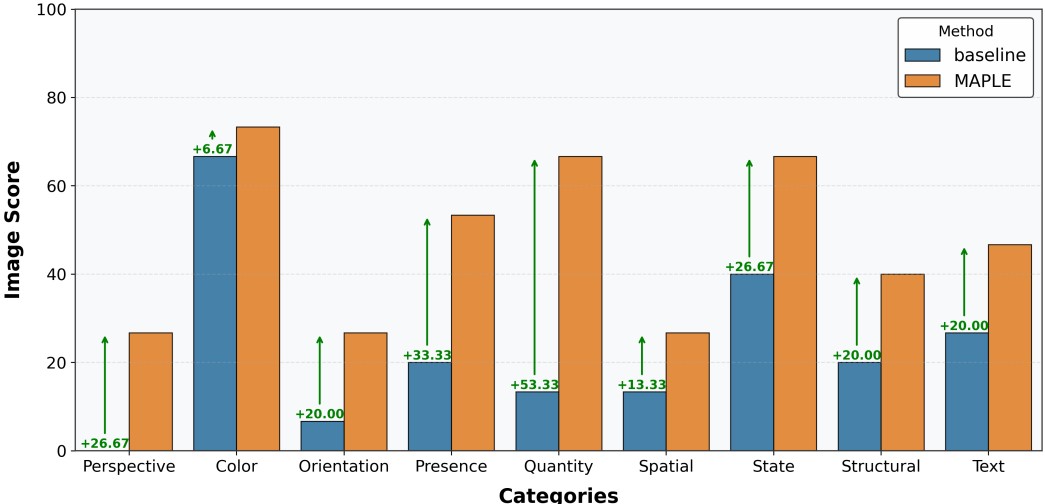

Figure 7: **Performance Breakdown Comparison on the MMVP dataset.**

a deduplication step on the globally gathered examples to ensure uniqueness before computing the final contrastive loss.

## C.2 Impact of Balancing $\mathcal{L}_{\mathbf{RPA}}$ with $\mathcal{L}_{\mathbf{contrast}}$

Our full method combines the targeted $\mathcal{L}_{\mathrm{RPA}}$ with the general $\mathcal{L}_{\mathrm{contrast}}$. We analyze the impact of the balancing weight $\lambda\mathcal{L}_{\mathrm{RPA}} + (1 - \lambda)\mathcal{L}_{\mathrm{contrast}}$. Figure 5 illustrates performance trajectories as $\lambda$ changes from 0 ($\mathcal{L}_{\mathrm{RPA}}$ only) to 1 ($\mathcal{L}_{\mathrm{contrast}}$ only). Interestingly, as $\lambda$ increases, general retrieval abilities slowly get worse, while fine-grained discrimination shows clear improvement. This opposite relationship highlights a basic trade-off in our approach. Finding the right value of $\lambda$ is important—it's a balance between general general retrieval and fine-grained retrieval. Also, our experiments show that $\mathcal{L}_{\mathrm{RPA\text{-}Listwise}}$ works much better than its pairwise version, with this advantage becoming more obvious at higher $\lambda$ values.

## C.3 Impact of the strategy of sampling captions

For our caption generation approach, we generate comparative descriptions between image pairs. Specifically, for each anchor image and its top-3 hard negative images, we construct six distinct image pairs by sampling without replacement from the four images. To ensure diversity in the captions, we employ a generation temperature of 0.7 and generate three captions per pair. This process yields a

Table 8: **Sampling Strategy Comparison on Generated Captions.** Best results are in **bold**. These experiments are conducted $\mathcal{L}_{\text{contrast}} + \mathcal{L}_{\text{RPA-Listwise}}$.

| Strategy | General Retrieval (R@1) | | | | Fine-grained Retrieval | | | |
| | COCO | | Flickr30k | | Winoground | | NaturalBench | |
| | Text | Image | Text | Image | Text | Image | Text | Image |
|---|---|---|---|---|---|---|---|---|
| A | 71.2 | 57.6 | 91.0 | 83.9 | 49.0 | 27.0 | 68.8 | 70.0 |
| B | 71.4 | 58.0 | 91.4 | 84.3 | 49.4 | 27.5 | **69.2** | 70.4 |
| C | **71.9** | **58.6** | **92.0** | **84.9** | **51.0** | **28.2** | **69.2** | **71.2** |

total of 18 comparative descriptions, with each image appearing in six different captions, resulting in a rich textual training corpus.

To understand the impact of different caption sampling strategies, we evaluate the following three approaches for each image. Strategy A uses only the first caption from each of the six generated caption sets, without any sampling. Strategy B randomly samples one caption from the first three generated captions per pair, without additional regeneration. Strategy C randomly samples from all six generated captions. As illustrated in Table 8, we find that increasing caption diversity gradually improves model performance. As for more sophisticated caption generation processes (e.g., leveraging multiple MLLMs to generate comparative descriptions and mixing them together), we leave them for future work.

### C.4 Detailed Performance Analysis on MMVP and BiVLC Datasets

While Table 3 provides overall metrics across different fine-grained datasets, we further analyze the improvements across specific visual patterns. The MMVP dataset categorizes visual tasks into 9 distinct patterns. As shown in Figure 7, the baseline demonstrates strong discriminative capabilities for Color and State patterns, but performs considerably worse on other patterns. Our MAPLE approach delivers significant improvements on these challenging patterns. Additionally, we compare the baseline with MAPLE across 3 categories in the BiVLC dataset, where each category represents a different dimension of variation between image-text pairs. Figure 8 reveals that MAPLE consistently improves upon baseline performance, with particularly notable gains in the Swap category.

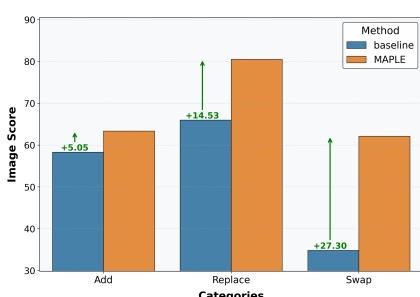

Figure 8: **Performance Breakdown Comparison on the BiVLC dataset.**

## D Related Work

**Cross-Modal Retrieval and the Modality Gap.** Cross-modal retrieval aims to establish correspondences between visual and textual information in a shared embedding space for effective search [39]. While contrastive learning approaches [1, 40, 41] have advanced the field by training separate encoders on paired datasets, they consistently face challenges from the modality gap [2–4, 42, 43]. Our work introduces a unified metric that quantifies this gap across different architectures, showing that MLLMs inherently possess strong cross-modal alignment capabilities. Recent MLLM-based retrieval models [8, 10, 9, 21, 44] have reduced this gap through unified architectures, but still rely on relatively simple alignment strategies that limit their effectiveness in tasks requiring nuanced cross-modal understanding.

**Preference Learning and Direct Preference Optimization.** Preference learning has become a central technique in fine-tuning large language models, particularly in alignment with human feedback [45, 46]. Direct Preference Optimization (DPO) [11] has emerged as a principled alternative to reinforcement learning-based methods, offering a stable, reward-free approach for modeling preferences. While DPO has shown success in natural language domains, its application to cross-modal representation learning remains underexplored. In this work, we extend DPO to retrieval-based

settings and propose a reinforcement learning framework that transfers the fine-grained modality alignment priors of MLLMs into the cross-modal representations learned by an MLLM-based retriever.

# E  Visualization of Retrieval Results

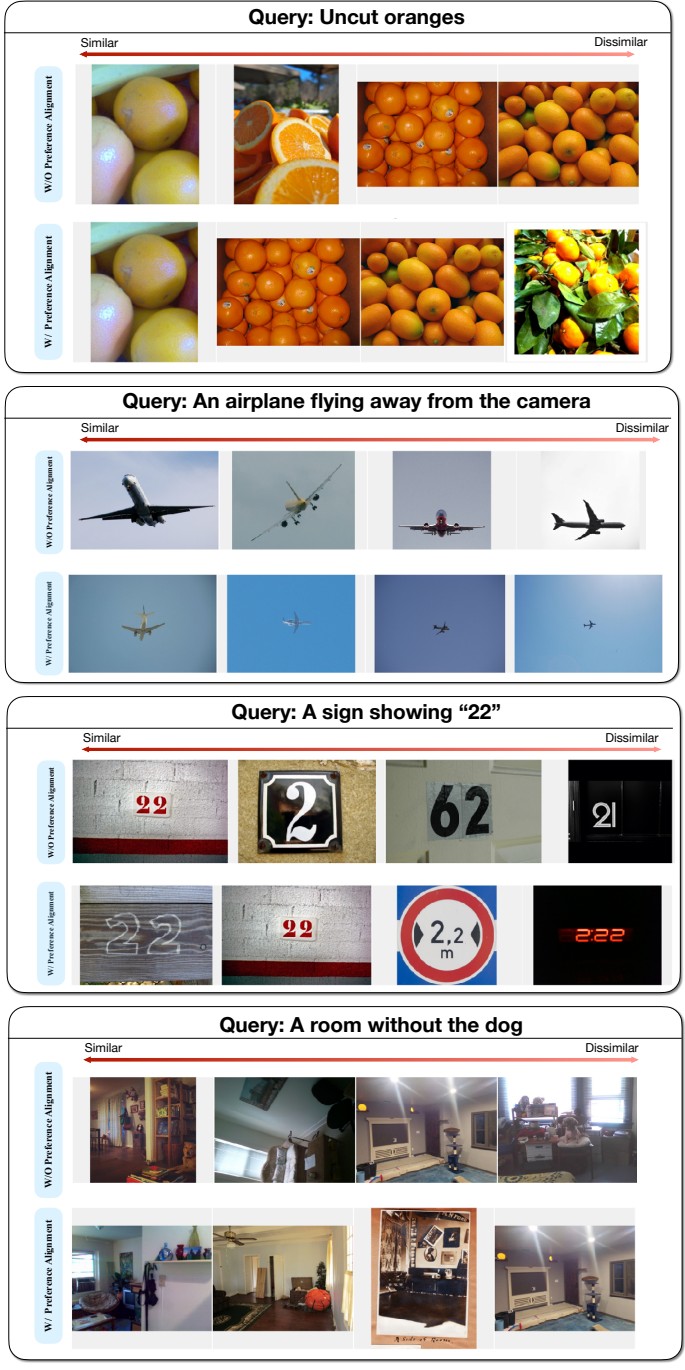

Figure 9: **Visualization Comparison.** The retrieved images are sorted by similarity scores. The first row shows retrieval results without preference alignment, while the second row shows results with preference alignment. (Best viewed in color and when zoomed in)

