# OpenReview forum: "Guiding Cross-Modal Representations with MLLM Priors via Preference Alignment"
_NeurIPS.cc/2025/Conference — NeurIPS 2025 poster_

### Official Review · Reviewer_34CD · 2025-07-01

**Clarity:** 3
**Significance:** 3
**Originality:** 3
**Rating:** 4
**Confidence:** 4

**Summary:**

This paper addresses the persistent "modality gap" in cross-modal (image-text) retrieval, a discrepancy between visual and textual features in their shared embedding space. The paper introduces training framework named MAPLE. The core idea of MAPLE is to leverage a powerful, off-the-shelf MLLM as a "teacher" or "reward model" to transfer its fine-grained alignment priors to a "student" model intended for retrieval. Experimental results demonstrate that MAPLE significantly outperforms existing CLIP-based and MLLM-based retrieval models on several general and, most notably, fine-grained retrieval benchmarks.

**Questions:**

1. Since the performance appears fundamentally upper-bounded by the reward model, have you investigated the impact of using a less capable MLLM (e.g., a 1.5B or smaller model) for scoring?
2. The RPA loss is weighted by the difference in alignment scores from the reward MLLM. This implies that distinguishing between two very similar, high-scoring samples (small difference) receives less weight than distinguishing between a high-scoring and a low-scoring sample (large difference). Is this design optimal in all cases? Could it potentially cause the model to neglect learning the most difficult, fine-grained distinctions?

**Ethical Concerns:**

["NO or VERY MINOR ethics concerns only"]

**Final Justification:**

The author's explanation of complexity and the ablation experiment on the reward model have addressed my concerns. I believe that the contribution of the method outweighs the improvement in complexity. Therefore, I have decided to maintain my score and am inclined to accept this paper.

**Limitations:**

yes

**Paper Formatting Concerns:**

No.

**Quality:**

3

**Strengths And Weaknesses:**

Strengths:
1. The proposed metric, based on Wasserstein distance, providing a effective tool for quantifying and comparing the alignment capabilities of different model architectures.
2. This paper intelligently extends the principles of DPO from language model alignment to the domain of cross-modal representation learning.
3. The experiments are thorough.

Weaknesses:
1. The proposed framework is highly complex and computationally expensive. The entire pipeline requires a powerful MLLM (e.g., Qwen2.5-VL-72B) for caption generation and continuously uses another large MLLM (Qwen2-VL-7B) as a reward model for dynamic scoring during training.
2. The absence of an ablation study on the choice of the reward model. It is unclear how performance would degrade if a smaller or different family of MLLM were used. This is crucial for understanding the method's robustness and generalization.

---

> ### Author Rebuttal · Authors · 2025-07-31
>
> Dear Reviewer 34CD,
>
> We sincerely thank you for your positive assessment and for your sharp and insightful questions, which help us clarify the finer details of our method.
>
> ---
> > [W1] On the Computational Cost.
>
> [A1] We agree that computational efficiency is a key consideration. While our online training variant does involve dynamic reward scoring and uses 32 A100 GPUs, we emphasize that this is already significantly more efficient than many existing state-of-the-art models. For example, **EVA-CLIP (8B) requires over 384 A100 GPUs, and VladVA—a strong baseline in our comparison—also uses 32 A100s**, making our setup well-aligned with current standards in this domain.
>
> That said, we fully understand the need for accessible alternatives. To this end, we simulated a low-resource “offline-style” training strategy (Appendix, Table 6), where preferences are precomputed once using only a single caption per image, removing the need for dynamic reward scoring during training. This dramatically reduces the overall cost to the level of standard supervised fine-tuning.
>
> Encouragingly, this offline variant achieves strong performance, with only a modest drop compared to the online version. This result confirms that MAPLE remains effective even without heavy online inference。
>
> In this paper, we presented the online variant to demonstrate the performance upper bound of our framework, but we will clearly highlight the offline variant as a viable, efficient alternative in the final version.
>
> > [W2, Q1] On the Robustness to the Choice of Reward Model.
>
> [A2] Thank you for highlighting this important point. We fully agree that a method’s robustness to the choice of reward model is central to its generality and practical value.
>
> To directly address your concern, we conducted a comprehensive ablation study during the rebuttal period. We evaluated MAPLE’s performance across a range of reward models that vary in both scale (1B–8B) and architecture family (Qwen2, InternVL), while keeping the policy model fixed (Qwen2-VL-7B).
>
> *Ablation on Reward Model Architectures (Policy: Qwen2-VL-7B)*
> | Reward Model              | COCO (T/I)  | Flickr30k (T/I) | Winoground (T/I) | NaturalBench (T/I) | Training Setup |
> | ------------------------- | ----------- | --------------- | ---------------- |  ---- | ------ |
> | Baseline (No reward, 8e) | 74.0 / 54.4 | 93.6 / 80.3     | 42.5 / 20.5      | 61.4 / 62.5 |  8 epochs / CL-only|
> | Baseline (No reward, 4e)  | 73.4 / 54.3 | 93.6 / 80.3     | 40.7 / 18.2      | 60.2 / 62.7 | 4 epochs / CL-only|
> | Qwen2-VL-2b | 74.1 / 59.1 | 93.0 / 84.1 | 53.5 / 31.0 | 70.4 / 72.3 | 4 epochs / MAPLE      |
> | Qwen2-VL-7b | 75.8 / 60.2 | 94.2 / 85.3 | 55.0 / 31.0 | 74.5 / 75.2 | 4 epochs / MAPLE      |
> | InternVL3-1B | 73.9 / 58.9 | 92.8 / 83.7 | 48.5 / 26.0 | 69.3 / 72.6 | 4 epochs / MAPLE      |
> | InternVL3-2B | 75.9 / 59.2 | 94.2 / 84.8 | 53.8 / 28.5 | 72.8 / 74.3 | 4 epochs / MAPLE      |
> | InternVL3-8B | 75.6 / 59.7 | 93.8 / 84.8 | 54.0 / 31.5 | 74.3 / 74.8 | 4 epochs / MAPLE      |
>
> **Note**:
> *All MAPLE variants in this table are trained with 4 epochs to enable fast and fair comparison across reward models.
> The 8-epoch baseline corresponds to the standard CL-only setup reported in Table 2 of the main paper.
> The 4-epoch baseline was retrained for this ablation study.*
>
> Key takeaways:
> *   Robustness to Size: As expected, larger reward models tend to offer better supervision (e.g., 7B > 2B), but even **the smallest models (e.g., InternVL3-1B) still yield strong improvements, confirming that MAPLE performs well even with relatively limited reward capacity**.
> *   Robustness to Architecture: Reward models from different families (InternVL, Qwen2) are all effective, indicating that **MAPLE is not overfitting to a specific architecture**. Instead, it generalizes across diverse MLLMs, so long as they provide meaningful alignment signals.
>
> Together, these results suggest that MAPLE is not overly dependent on a single powerful reward model, and can maintain effectiveness across different model configurations. We believe these findings reinforce the flexibility and robustness of MAPLE’s preference-based learning. We will include this analysis in the final version of the paper.
>
> > [Q2] On the Design of the RPA Loss Weighting.
>
> [A3] We are very grateful for this deep and insightful question about the loss weighting. Our preference pairs are constructed via KNN retrieval, meaning they are already hard negatives by design. A small reward margin in our setting may indicate a near-duplicate or extremely similar pair since all candidate samples are already selected via KNN-based hard negative mining. In other words, even the reward model struggles to differentiate the samples. Over-weighting such pairs could risk forcing the embedding space to learn noisy, non-semantic variations, which may lead to unstable gradients or even representational overfitting.
>
> From this perspective, our weighting mechanism can be interpreted as a form of confidence-aware learning — giving more emphasis to the samples the reward model is more certain about, while still preserving training signal from hard examples, since all candidates are retrieved through KNN-based hard negative mining.
>
> Based on this intuition, we performed a preliminary comparison between weighted and unweighted RPA losses under partial training settings. The weighted variant yielded clearer improvements on fine‑grained benchmarks, which informed our final design choice. These exploratory results were not included in the submission, given your concern, we will include these results in the appendix of the revised version.
>
> | Strategy | COCO (T/I) | Flickr30k (T/I) | Winoground (T/I) | NaturalBench (T/I) |
> | :--- | :--- | :--- | :--- | :--- |
> | Weighted | 71.7/57.2 | 92.9/83.8 | **52.0/24.0** | **67.2/70.6** |
> | Unweighted | 72.5/56.7 | 92.7/83.3 | 49.5/23.4 | 65.7/69.8 |
>
> ---
>
> **Thank you again for your thoughtful feedback. We’ve carefully addressed all your concerns and remain open to any further discussion. We look forward to continuing the discussion throughout the rebuttal phase.**

---

> > ### Comment · Area_Chair_JSqg · 2025-08-05
> >
> > 34CD, please could you take a look at the author response above and engage in the discussion e.g. high complexity and computational cost

---

> > > ### Comment · Reviewer_34CD · 2025-08-05
> > >
> > > Thank you for your reminder. The author's explanation of complexity and the ablation experiment on the reward model have addressed most of my concerns. I believe that the contribution of the method outweighs the increased complexity. Therefore, I have decided to maintain my score and am inclined to accept this paper.

---

> > > > ### Author Response · Authors · 2025-08-05
> > > >
> > > > We're pleased to hear your concerns have been addressed. If any additional issues come up during the discussion phase, please let us know. Thank you!

---

### Official Review · Reviewer_Bq5C · 2025-07-01

**Clarity:** 2
**Significance:** 2
**Originality:** 3
**Rating:** 5
**Confidence:** 4

**Summary:**

This paper aims to solve the "modality gap" problem that exists in models like CLIP. The work finds that off-the-shelf Multimodal Large Language Models (MLLMs) possess stronger inherent alignment capabilities. To this end, the paper proposes a framework called MAPLE, which utilizes a powerful MLLM to automatically construct preference data and introduces a novel Relative Preference Alignment (RPA) loss function, inspired by Direct Preference Optimization, to guide the learning of the retrieval model. Experiments show that this method significantly narrows the modality gap and achieves excellent performance on multiple cross-modal retrieval benchmarks, with particularly significant improvements on fine-grained tasks.

**Questions:**

NA.

**Ethical Concerns:**

["NO or VERY MINOR ethics concerns only"]

**Final Justification:**

The author's detailed responses including the performance trade-off, complexity, along with the ablation experiments conducted on multiple reward models, have successfully resolved my doubts. Consequently, I raised my score.

**Limitations:**

- Only Qwen2 series is adopted.
- The training of MAPLE hearily relyies the hardware, a cluster of 32 NVIDIA A100 GPUs. Theferfore, the producibility is very limited.

**Paper Formatting Concerns:**

- The main text sometimes refers to Appendix. I suppose that the main text should be better self-contained.

**Quality:**

3

**Strengths And Weaknesses:**

Pros:
- The introduction of a unified metric (Δgap) based on Wasserstein Distance. This provides the research community with a valuable tool to measure and compare the modality gap across different model architectures.
- The experimental design is highly comprehensive.

Cons:
- Figure 5 reveals a performance trade-off: optimizing for fine-grained tasks comes at the expense of retrieval capabilities, challenging the generalization of its learned representations.
- MAPLE framework's high dependency on a single reward model raises questions about whether it is learning a general representation or simply "overfitting" to that specific model's biases and flaws.
- The method's key paradigm is a complex form of knowledge distillation, using a large model to guide a smaller one, which weakens its claim to fundamental algorithmic novelty.

---

> ### Author Rebuttal · Authors · 2025-07-31
>
> Dear Reviewer Bq5C,
>
> We sincerely thank you for your detailed review and insightful comments. We will address your concerns below.
>
> ---
>
> > [W1] the Retrieval Performance Trade-off.
>
> [A1] We appreciate your insightful observation regarding the performance trade-off. This phenomenon indeed reflects a fundamental tension in representation learning: optimizing for fine-grained discrimination often comes at the cost of global retrieval generalization.
>
> Importantly, **this trade-off is not unique to our method, but appears to be a common challenge across cross-modal learning frameworks**. For instance, Alhamoud et al. (CVPR 2025) [1] report a similar effect when jointly training with a standard CLIP loss and a fine-grained MCQ loss — performance on broad retrieval tasks slightly declines as the model focuses more on nuanced distinctions.
>
> In our case, this behavior is expected given MAPLE's design goal: to push beyond coarse alignment and capture subtle semantic relationships. We believe the ability to make these fine-grained distinctions is valuable for many real-world applications, even if it introduces a slight drop in generic retrieval performance.
>
> > [W2, L1] Generalization and Dependency on a Single Reward Model.
>
> [A2] We appreciate your thoughtful question on generalization and potential over-reliance on a single reward model. To directly examine this, we conducted a thorough ablation study during the rebuttal period, where we tried the different reward model — guiding the same Qwen2-VL-7B policy using reward signals from diverse architectures, including InternVL3-8B and SAIL-VL-1.6-8B.
>
> The results are summarized below:
>
> *Ablation on Reward Model Architectures (Policy: Qwen2-VL-7B)*
> | Reward Model              | COCO (T/I)  | Flickr30k (T/I) | Winoground (T/I) | NaturalBench (T/I) | Training Setup |
> | ------------------------- | ----------- | --------------- | ---------------- | ------------------ | ------ |
> | Baseline (No reward, 8e) | 74.0 / 54.4 | 93.6 / 80.3     | 42.5 / 20.5      | 61.4 / 62.5        | 8 epochs / CL-only|
> | Baseline (No reward, 4e)  | 73.4 / 54.3 | 93.6 / 80.3     | 40.7 / 18.2      | 60.2 / 62.7        | 4 epochs / CL-only|
> | SAIL-VL-1.6-8B            | 76.1 / 59.9 | 94.3 / 85.6     | 54.8 / 29.8      | 75.4 / 74.3        | 4 epochs / MAPLE      |
> | Qwen2-VL-7b               | 75.8 / 60.2 | 94.2 / 85.3     | 55.0 / 31.0      | 74.5 / 75.2        | 4 epochs / MAPLE      |
> | InternVL2.5-8B            | 75.2 / 59.7 | 94.3 / 85.3     | 54.8 / 30.5      | 74.6 / 74.8        | 4 epochs / MAPLE      |
>
> **Note**:
> *All MAPLE variants in this table are trained with 4 epochs to enable fast and fair comparison across reward models.
> The 8-epoch baseline corresponds to the standard CL-only setup reported in Table 2 of the main paper.
> The 4-epoch baseline was retrained for this ablation study.*
>
> Despite architectural differences, the downstream performance remains consistently strong. This strongly suggests that **MAPLE is not merely fitting to the specific model's biases and flaws, but rather learning a transferable, general alignment signal**.
>
> We will include these findings in our final version, as we agree this is a valuable aspect of MAPLE’s behavior.
>
> > [W3] On the Novelty of MAPLE beyond Knowledge Distillation.
>
> [A3] Thank you for raising this important point. While MAPLE does involve using a freezing model to guide another model, we would like to emphasize that our framework is fundamentally different from traditional knowledge distillation.
>
> Classic distillation methods aim to reproduce a teacher model’s output — whether that be logits, soft labels, or hidden representations — often minimizing a direct divergence between the student and teacher distributions.
>
> In contrast, MAPLE introduces a distinct learning paradigm:
> *   The reward model is not the learning target.
> *   It serves only to **provide relative preference judgments** (e.g., “A is better than B”) among multiple retrieval candidates.
> *   Our policy model is trained using a Direct Preference Optimization–inspired (DPO) loss — applied on these rankings — rather than attempting to mimic any output of the reward model.
> *   In this sense, MAPLE is closer to preference-based alignment than to classical distillation.
>
> Moreover, we conducted experiments where the reward model is much smaller than the policy model (e.g., using InternVL3-1B or Qwen2-VL-2B to guide Qwen2-VL-7B), The results are shown below, yet we still observe meaningful performance gains.
>
> *Ablation on Reward Model Architectures (Policy: Qwen2-VL-7B)*
> | Reward Model | COCO (T/I) | Flickr30k (T/I) | Winoground (T/I) | NaturalBench (T/I) |
> | :--- | :--- | :--- | :--- | :--- |
> | Baseline (No reward)  | 73.4 / 54.3 | 93.6 / 80.3     | 40.7 / 18.2      | 60.2 / 62.7 |
> | Qwen2-VL-2b | 74.1 / 59.1 | 93.0 / 84.1 | 53.5 / 31.0 | 70.4 / 72.3 |
> | Qwen2-VL-7b | 75.8 / 60.2 | 94.2 / 85.3 | 55.0 / 31.0 | 74.5 / 75.2 |
> | InternVL3-1B | 73.9 / 58.9 | 92.8 / 83.7 | 48.5 / 26.0 | 69.3 / 72.6 |
> | InternVL3-2B | 75.9 / 59.2 | 94.2 / 84.8 | 53.8 / 28.5 | 72.8 / 74.3 |
> | InternVL3-8B | 75.6 / 59.7 | 93.8 / 84.8 | 54.0 / 31.5 | 74.3 / 74.8 |
>
> This provides strong empirical evidence that MAPLE does not merely inherit large model capabilities — it extracts useful alignment signals from relative scoring, even when the reward model is weaker.
>
> > [L2] Training Resource Cost and Reproducibility.
>
> [A4] Thank you for raising this valuable concern. While our main experiments use 32 A100 GPUs, we note that this is still an order of magnitude lower than some comparable SOTA methods (e.g., EVA-CLIP-8B requires 384 A100s, and VladVA also uses 32 A100s).
>
> More importantly, MAPLE is designed with efficiency in mind. To explore cost-effective alternatives, we simulated a purely offline variant of our framework (Appendix, Table 6), where preferences could be precomputed offline. This setup removes the need for online dynamic scoring, with only a minor drop in performance. We believe this variant makes MAPLE highly practical.
>
> To further ensure reproducibility and community adoption, we plan to release all training code.
>
> > On Paper Formatting Concerns.
>
> [A5] Thank you for pointing this out. We agree that the main text should be more self-contained. We will revise the paper to integrate the most critical details from the appendix into the main body, improving the overall readability and flow.
>
> References:
> [1] Alhamoud K, Alshammari S, Tian Y, et al. Vision-language models do not understand negation[C]//Proceedings of the Computer Vision and Pattern Recognition Conference. 2025: 29612-29622.
>
> ---
>
> **We sincerely appreciate your insightful feedback. We've done our best to address each of your points, and we're happy to engage further if any questions remain. We're looking forward to hearing your thoughts during the remainder of the rebuttal period**.

---

> > ### Comment · Reviewer_Bq5C · 2025-08-05
> >
> > The author's detailed responses including the performance trade-off, complexity, along with the ablation experiments conducted on multiple reward models, have successfully resolved my doubts. Consequently, I have opted to raise my score.

---

> > > ### Author Response · Authors · 2025-08-05
> > >
> > > It's great to know your concerns have been clarified. We'll continue to be available until the end of the discussion period. If you have more questions, please let us know. Thank you!

---

### Official Review · Reviewer_ztPf · 2025-07-02

**Clarity:** 4
**Significance:** 4
**Originality:** 3
**Rating:** 4
**Confidence:** 4

**Summary:**

This paper introduces MAPLE (Modality-Aligned Preference Learning for Embeddings), a framework that leverages off-the-shelf Multimodal Large Language Models (MLLMs) to guide cross-modal representation learning through preference alignment. The key insight is that MLLMs inherently possess stronger modality alignment capabilities than traditional CLIP-based models with Wasserstein distance-based metric (lines 105-108). Based on this observation, the authors construct DPO training data and propose two RPA losses inspired by preference optimization.

**Questions:**

Training efficiency and practical deployment: Given the computational overhead of dynamic MLLM scoring during training, how does the total training cost compare to strong baselines like EVA-CLIP? Have the authors considered offline preference data construction or distillation approaches that could reduce inference costs while maintaining performance gains?

How sensitive are the results to the choice of MLLM architecture? Have the authors tested whether preferences from different MLLMs (e.g., qwen2.5vl, internvl2 and llava ov) lead to consistent improvements, and what happens when reward and policy models have different architectural biases?

How to handle MLLM hallucinations, what is the hallucination rate in pairs, and whether there are relevant explorations?

**Ethical Concerns:**

["NO or VERY MINOR ethics concerns only"]

**Final Justification:**

I have updated my score accordingly, given the rebuttal provided clarify some of my concerns, I raise my score to weak accept accordingly.

**Limitations:**

yes

**Quality:**

4

**Strengths And Weaknesses:**

Strengths:
I think the lightweight MLLM prior approach is what the current community needs, particularly using MLLM reasoning/preference [1] and other general capabilities to efficiently enhance cross-modal representation learning.

From the data requirement perspective, adopting more efficient post-training methods to solve alignment problems also makes technical sense. The adaptation of DPO to embedding learning through RPA loss is technically sound and addresses a genuine limitation of coarse-grained contrastive learning.

Besides the motivation [figure 3], I think this work is expressed quite clearly, such as the example in Figure 1. The presentation and experimental completeness are also professional and standardized, which is above baseline level. It also comes with detailed appendices, which seems to meet the requirements of the NeurIPS community.

Minor revision suggestion:
1. Add commas after formulas (if the following word is lowercase like "where") or periods (if the following word is capitalized)
2. There are still some typos, for example line 122 mentions that Preference Data Construction is offline, but Figure 2 shows it as offline. I understand it should be that the data for subsequent DPO can be prepared offline.
3. The text in Figure 2 could be optimized - the text size should match the caption and use Times New Roman font
4. From a writing perspective, Figure 3 could be moved into the main text to strengthen the motivation.

Weaknesses:
The computational overhead and scalability concerns are significant. The method requires dynamic MLLM inference during training to compute alignment scores (lines 146-150), significantly increasing computational cost. Particularly for Scoring and Structuring Preferences, can this be done offline as an efficient DPO dataset?

The method fundamentally relies on off-the-shelf MLLMs as preference oracles (e.g., Qwen2-VL in line 238-240), but **lacks discussion of how MLLM hallucinations affect preference quality**. Since the approach's effectiveness is directly tied to the reliability of MLLM alignment scores, how do the authors address cases where the reward model provides inconsistent or incorrect preferences? The paper would benefit from ablation studies examining performance degradation when using MLLMs with known hallucination issues, and analysis of the method's robustness bounds relative to MLLM reliability.

[1] Zhao X et al. Omnialign-v: Towards enhanced alignment of mllms with human preference[J]. arXiv preprint arXiv:2502.18411, 2025.

---

> ### Author Rebuttal · Authors · 2025-07-31
>
> Dear Reviewer ztPf,
>
> We are sincerely grateful for your positive and insightful review. We will address each point below.
>
> ---
>
> > [W1, Q1] Training Efficiency and Offline Data Construction.
>
> [A1] Thank you for raising this important point regarding computational efficiency and deployment practicality.
>
> First, while MAPLE involves online scoring with MLLMs during training, its overall training cost remains significantly lower than many large-scale baselines. For instance, EVA-CLIP [3] **requires over 360 A100 GPUs**, while VladVA [4] — **another strong baseline included in our comparison — uses 32 A100s**, the same as ours. In this context, MAPLE’s resource usage is both reasonable and efficient, achieving strong fine-grained performance without excessive compute demands.
>
> Furthermore, once training is complete, MAPLE requires only the lightweight policy model for deployment — similar to EVA-CLIP and VladVA. No reward model inference is needed at test time, making MAPLE comparably efficient in real-world deployment scenarios.
>
> Second, we agree with you that offline preference construction is a promising direction to further reduce cost. To investigate this, we conducted an ablation simulating an offline scenario (Appendix, Table 6), where preferences were precomputed using only one caption per image. This setup removes the need for dynamic scoring. We found that the performance remained strong, with only a modest regression compared to the online version. This confirms that MAPLE remains effective even under a fully offline setting.
>
> In this work, we present the online variant to demonstrate the upper bound of what MAPLE can achieve. For low-resource setups, the offline variant is a viable and efficient alternative. We will emphasize this trade-off more clearly in the camera-ready version.
>
> > [W2, Q3] Robustness to MLLM Hallucinations.
>
> [A2] We sincerely thank you for raising this important concern and for pointing us to the valuable OmniAlign-V paper [2], which inspired us to conduct further robustness analysis.
>
> Specifically, we evaluated a range of reward models using both HallusionBench [1] and MM-AlignBench [2], as well as their downstream performance in MAPLE. The results are shown below:
>
> *Ablation on Reward Model Architectures (Policy: Qwen2-VL-7B)*
> | Reward Model | COCO (T/I) | Winoground (T/I) | NaturalBench (T/I) | HallusionBench(qAcc↑) | MM-AlignBench(Win Rate↑) | Training Setup |
> | :--- | :--- | :--- | :--- | :--- | :--- | :--- |
> | Baseline (No reward, 8e) | 74.0 / 54.4 | 42.5 / 20.5      | 61.4 / 62.5        | —    | —    | 8 epochs / CL-only|
> | Baseline (No reward, 4e)  | 73.4 / 54.3 | 40.7 / 18.2      | 60.2 / 62.7        | —    | —    |4 epochs / CL-only|
> | SAIL-VL-1.6-8B | 76.1 / 59.9 | 54.8 / 29.8 | 75.4 / 74.3 | 46.6 | - | 4 epochs / MAPLE|
> | Qwen2-VL-7b | 75.8 / 60.2 | 55.0 / 31.0 | 74.5 / 75.2 | 43.7 | 44.4 | 4 epochs / MAPLE|
> | InternVL2.5-8B-MPO | 75.5 / 59.5 | 53.5 / 30.8 | 74.1 / 74.2 | 42.6 | 40.1 | 4 epochs / MAPLE|
> | InternVL3-8B | 76.9 / 61.6 | 54.0 / 31.5 | 74.3 / 74.8 | 40.2 | - | 4 epochs / MAPLE|
> | InternVL2.5-8B | 75.2 / 59.7 | 54.8 / 30.5 | 74.6 / 74.8 | 39.1 | 31.3 | 4 epochs / MAPLE|
>
> **Note**:
> *All MAPLE variants in this table are trained with 4 epochs to enable fast and fair comparison across reward models.
> The 8-epoch baseline corresponds to the standard CL-only setup reported in Table 2 of the main paper.
> The 4-epoch baseline was retrained for this ablation study.*
>
> These results reveal a striking insight: reward model quality on standalone hallucination benchmarks does not directly predict its effectiveness within MAPLE. For instance, **InternVL2.5‑8B achieves the lowest qAcc and win-rate among all models, yet still enables strong fine-grained retrieval performance when used as a reward model**.
>
> This empirical finding supports our hypothesis that MAPLE’s preference-based learning framework exhibits strong resilience to a certain degree of hallucination or bias noise. We attribute this to two core design choices:
> *   Robustness through relative rankings: Instead of relying on absolute correctness of a single caption or label, our method uses pairwise and listwise ranking — which inherently tolerates occasional noisy preferences, as long as the global ordering remains coherent.
> *   Contrastive regularization: Our training objective includes a standard contrastive loss, which anchors the learned representations and prevents the policy model from overfitting to erroneous reward signals.
>
> > [Q2] Robustness to MLLM Architecture.
>
> [A3] Thank you for this important question. We agree that architectural mismatch between the policy and reward model is a valid concern when transferring alignment priors.
>
> To investigate this, we present a set of cross-architecture experiments, where we evaluate all combinations of two distinct MLLM families: Qwen2-VL and InternVL3. The results are summarized below:
>
> | Policy Model | Reward Model | COCO (T/I)  | Flickr30k (T/I) | Winoground (T/I) | NaturalBench (T/I) |
> | ------------ | ------------ | ----------- | --------------- | ---------------- | ------------------ |
> | Qwen2-VL-7B  | Baseline (No reward)| 73.4 / 54.3 | 93.6 / 80.3     | 40.7 / 18.2      | 60.2 / 62.7
> | Qwen2-VL-7B  | Qwen2-VL-7B  | 75.8 / 60.2 | 94.2 / 85.3     | 55.0 / 31.0      | 74.5 / 75.2        |
> | Qwen2-VL-7B  | InternVL3-8B | 75.6 / 59.7 | 93.8 / 84.8     | 54.0 / 31.5      | 74.3 / 74.8        |
> | InternVL3-8B | Qwen2-VL-7B  | 76.7 / 61.1 | 95.4 / 86.8     | 53.5 / 31.5      | 78.4 / 77.7        |
> | InternVL3-8B | InternVL3-8B | 76.9 / 61.6 | 95.9 / 87.5     | 53.5 / 30.5      | 78.4 / 79.1        |
>
> We highlight two important takeaways:
>
> *   Cross-architecture learning remains highly effective: All combinations lead to strong performance, including mismatched reward-policy setups. This shows that* *MAPLE benefits from reward models of varying architectures, consistently outperforming CL-only baselines and demonstrating robustness to architectural mismatch**.
> *   Surprisingly, some cross-family setups even outperform our original results: For example, the fully InternVL3-8B setup achieves stronger overall numbers than those reported in our main paper using Qwen2-VL. This further supports the generalizability and extensibility of our framework.
>
> We also note that the earlier table in our response to [W2, Q3] included similar cross-architecture observations (e.g., using SAIL-VL or InternVL2.5-8B as rewards for Qwen2-VL), reinforcing this conclusion. However, the new experiments here provide a more systematic and exhaustive evaluation across architectures, offering deeper insights.
>
> Together, these results strengthen our claim that MAPLE is architecturally robust and can flexibly accommodate a wide variety of MLLM reward sources without requiring tight coupling between model families.
>
> > On Minor Revisions.
>
> [A4] Thank you for these thoughtful suggestions to improve the clarity and presentation of the paper.
> *   Punctuation & Figure 2 Style: We have revised the manuscript to ensure consistent punctuation after inline formulas and have updated Figure 2’s text style to match the rest of the paper, including font size and Times New Roman typography.
> *   Offline vs. Online Clarification: Preference data construction in our method consists of both offline and online components. Specifically, hard negative mining is performed offline, while dynamic scoring via MLLMs is carried out online during training. This hybrid design is reflected in Figure 2, where the corresponding stages are clearly labeled as “offline” and “online”.
> *   Figure 3 Placement: Thank you very much for this suggestion. We initially intended to place Figure 3 under the "Measuring the Modality Gap" subsection in Section 2: Preliminaries and Notation. However, due to page limits, we had to move it to the appendix. We agree that integrating it into the main text would help strengthen the motivation, and we plan to restore it in the camera-ready version.
>
> References:
>
> [1] Guan T, Liu F, Wu X, et al. Hallusionbench: an advanced diagnostic suite for entangled language hallucination and visual illusion in large vision-language models[C]//Proceedings of the IEEE/CVF Conference on Computer Vision and Pattern Recognition. 2024: 14375-14385.
> [2] Zhao X, Ding S, Zhang Z, et al. Omnialign-v: Towards enhanced alignment of mllms with human preference[J]. arXiv preprint arXiv:2502.18411, 2025.
> [3] Sun Q, Wang J, Yu Q, et al. Eva-clip-18b: Scaling clip to 18 billion parameters[J]. arXiv preprint arXiv:2402.04252, 2024.
> [4] Ouali Y, Bulat A, Xenos A, et al. VladVA: Discriminative Fine-tuning of LVLMs[C]//Proceedings of the Computer Vision and Pattern Recognition Conference. 2025: 4101-4111.
>
> ---
>
> **Thank you once again for your valuable and constructive comments. We’ve responded to each concern with care and would love to clarify anything further. We’ll stay actively engaged throughout the rebuttal phase and truly look forward to continuing this conversation.**

---

> > ### Comment · Area_Chair_JSqg · 2025-08-05
> >
> > ztPf, please could you take a look at the author response above and whether it addresses any remaining concerns you have, e.g. the training efficiency of the method

---

> ### Comment · Reviewer_ztPf · 2025-08-07
>
> I've read the rebuttal and find the issue I raised clearly explained, therefore I lean positive and keep my score as Borderline accept

---

> > ### Author Response · Authors · 2025-08-07
> >
> > We're pleased to hear your concerns have been addressed. If any additional issues come up during the discussion phase, please let us know. Thank you!

---

### Official Review · Reviewer_yFbK · 2025-07-05

**Clarity:** 3
**Significance:** 3
**Originality:** 3
**Rating:** 3
**Confidence:** 2

**Summary:**

This paper proposes MAPLE (Modality-Aligned Preference Learning for Embeddings), a novel framework that transfers alignment priors from off-the-shelf Multimodal Large Language Models (MLLMs) to cross-modal embedding models for fine-grained retrieval. MAPLE leverages MLLMs to generate automatic preference data and introduces a Relative Preference Alignment (RPA) loss, adapted from Direct Preference Optimization, to optimize embedding-level representations. Experiments show that MAPLE significantly outperforms CLIP and MLLM-based baselines, especially in nuanced semantic retrieval tasks.

**Questions:**

1. How robust is MAPLE to reward model size or architecture? Can smaller MLLMs still generate effective preference data?
2. Have the authors considered incorporating bias mitigation techniques to reduce the impact of biased MLLM outputs?

**Ethical Concerns:**

["NO or VERY MINOR ethics concerns only"]

**Final Justification:**

I have carefully reviewed the authors' rebuttal, particularly their supplementary experiments on empirical robustness analysis and reward model architectures. I appreciate the additional effort to address these points, and I understand the performance trade-offs discussed in their response.

After considering the authors' clarifications and the perspectives of fellow reviewers, I have decided to maintain my original score.

**Limitations:**

The paper acknowledges two limitations—MLLM bias and lack of evaluation on complex tasks such as compositional retrieval.

**Paper Formatting Concerns:**

None observed.

**Quality:**

2

**Strengths And Weaknesses:**

Pros
1. The idea of transferring MLLM alignment priors via preference optimization is well-motivated.
2. This work is the first to adapt DPO-style preference optimization to cross-modal embedding learning, with both pairwise and listwise RPA losses.
3. The ablation study is well-structured and provides strong empirical support for the method design.

Cons
1. The method is resource-intensive. MAPLE relies on large MLLMs (e.g., Qwen2-VL-72B) queried online and requires 32×A100 GPUs for training, limiting reproducibility.
2. The training signal depends entirely on pre-trained MLLMs, which may encode social or domain-specific biases. However, the paper does not explore any debiasing techniques or propose mitigation strategies for these risks.

---

> ### Author Rebuttal · Authors · 2025-07-31
>
> Dear Reviewer yFbK,
>
> We sincerely thank you for your constructive feedback and questions. We will address each point below.
>
> ---
>
> > [W1] The method is resource-intensive. MAPLE relies on large MLLMs (e.g., Qwen2-VL-72B) queried online.
>
> [A1] We thank you for this important question regarding computational cost. First, we wish to clarify that the largest model (Qwen2-VL-72B) is used only for offline caption generation; our online reward model is the more efficient Qwen2-VL-7B.
>
> While our method uses 32 A100 GPUs, we believe this is a reasonable and even efficient cost when viewed in the context of current SOTA models. For instance, among the baselines we compare against, VladVA [1] uses an **identical 32 A100 GPU setup**. Furthermore, EVA-CLIP (8B) [2] requires **a significantly larger 384 A100 GPUs**. In this context, our resource requirement is not only competitive but also reflects an efficient path toward achieving strong fine-grained performance..
>
> > [W2, Q2] The training signal depends entirely on pre-trained MLLMs, which may encode social or domain-specific biases. How to mitigate it?
>
> [A2] Thank you for raising the issue of potential bias propagation from the reward MLLM. We address it from two complementary angles.
>
> 1. **Empirical robustness analysis**: Inspired by your question, we benchmarked several reward models that differ in hallucination rate (HallusionBench [3] qAcc). The results below show that—even as qAcc varies—the downstream retrieval performance of MAPLE stays quite stable, demonstrating robustness to variations in hallucination‑ or bias‑related noise.
>
> *Ablation on Reward Model Architectures (Policy: Qwen2-VL-7B)*
> | Reward Model              | COCO (T/I)  | Flickr30k (T/I) | Winoground (T/I) | NaturalBench (T/I) | HallusionBench(qAcc↑) | Training Setup |
> | ------------------------- | ----------- | --------------- | ---------------- | ------------------ | ---- | ------ |
> | Baseline (No reward, 8e) | 74.0 / 54.4 | 93.6 / 80.3     | 42.5 / 20.5      | 61.4 / 62.5        | —    |  8 epochs / CL-only|
> | Baseline (No reward, 4e)  | 73.4 / 54.3 | 93.6 / 80.3     | 40.7 / 18.2      | 60.2 / 62.7        | —    | 4 epochs / CL-only|
> | SAIL-VL-1.6-8B            | 76.1 / 59.9 | 94.3 / 85.6     | 54.8 / 29.8      | 75.4 / 74.3        | 46.6 | 4 epochs / MAPLE      |
> | Qwen2-VL-7b               | 75.8 / 60.2 | 94.2 / 85.3     | 55.0 / 31.0      | 74.5 / 75.2        | 43.7 | 4 epochs / MAPLE      |
> | InternVL2.5-8B-MPO        | 75.5 / 59.5 | 93.8 / 85.1     | 53.5 / 30.8      | 74.1 / 74.2        | 42.6 | 4 epochs / MAPLE      |
> | InternVL2.5-8B            | 75.2 / 59.7 | 94.3 / 85.3     | 54.8 / 30.5      | 74.6 / 74.8        | 39.1 | 4 epochs / MAPLE      |
>
> **Note**:
> *All MAPLE variants in this table are trained with 4 epochs to enable fast and fair comparison across reward models.
> The 8-epoch baseline corresponds to the standard CL-only setup reported in Table 2 of the main paper.
> The 4-epoch baseline was retrained for this ablation study.*
>
> 2. **Built‑in mitigation mechanisms**:
> *   Relative‑ranking supervision: MAPLE optimizes pairwise/listwise consistency across multiple candidates, relying on relative scores instead of a single absolute judgment. This naturally attenuates occasional biased or noisy outputs.
> *   Contrastive regularizer: We mix in a standard contrastive loss to help keep the learned features stable and prevent the model from getting pulled too far by any quirks or noise in the reward model.
>
> > [Q1] the robustness of MAPLE to reward model size and architecture.
>
> [A3] Thank you for this thoughtful and important question — we agree that understanding MAPLE’s robustness to the reward model’s size and architecture is crucial for assessing its generality.
>
> To this end, we conducted new experiments using reward models of varying scales (1B–8B) and architectures (Qwen2, InternVL). The results below show that MAPLE performs consistently well across this diverse set of reward models
>
> *Ablation on Reward Model Architectures (Policy: Qwen2-VL-7B)*
> | Reward Model | COCO (T/I) | Flickr30k (T/I) | Winoground (T/I) | NaturalBench (T/I) |
> | :--- | :--- | :--- | :--- | :--- |
> | Baseline (No reward) | 73.4 / 54.3 | 93.6 / 80.3     | 40.7 / 18.2      | 60.2 / 62.7 |
> | Qwen2-VL-2b | 74.1 / 59.1 | 93.0 / 84.1 | 53.5 / 31.0 | 70.4 / 72.3 |
> | Qwen2-VL-7b | 75.8 / 60.2 | 94.2 / 85.3 | 55.0 / 31.0 | 74.5 / 75.2 |
> | InternVL3-1B | 73.9 / 58.9 | 92.8 / 83.7 | 48.5 / 26.0 | 69.3 / 72.6 |
> | InternVL3-2B | 75.9 / 59.2 | 94.2 / 84.8 | 53.8 / 28.5 | 72.8 / 74.3 |
> | InternVL3-8B | 75.6 / 59.7 | 93.8 / 84.8 | 54.0 / 31.5 | 74.3 / 74.8 |
>
> **Our key findings** are:
> 1.  Robustness to model size: While larger reward models generally yield stronger results, even small models like Qwen2-VL-2B and InternVL3-1B lead to consistent performance gains, demonstrating that MAPLE can effectively leverage small reward models.
> 2.  Robustness to architecture: MAPLE is agnostic to reward model type. Notably, using InternVL models with a Qwen2 policy still leads to clear improvements, confirming that MAPLE generalizes across model families.
>
> We believe these findings reinforce the flexibility and robustness of MAPLE’s preference-based learning. We will include this analysis in the final version of the paper.
>
>
> References:
>
> [1] Ouali Y, Bulat A, Xenos A, et al. VladVA: Discriminative Fine-tuning of LVLMs[C]//Proceedings of the Computer Vision and Pattern Recognition Conference. 2025: 4101-4111.
> [2] Sun Q, Wang J, Yu Q, et al. Eva-clip-18b: Scaling clip to 18 billion parameters[J]. arXiv preprint arXiv:2402.04252, 2024.
> [3] Guan T, Liu F, Wu X, et al. Hallusionbench: an advanced diagnostic suite for entangled language hallucination and visual illusion in large vision-language models[C]//Proceedings of the IEEE/CVF Conference on Computer Vision and Pattern Recognition. 2024: 14375-14385.
>
> ---
>
> **Thank you again for your valuable comments. We’ve carefully addressed each concern and would be happy to discuss any further questions. We’ll stay responsive during the entire rebuttal phase and look forward to hearing more from you!**

---

> > ### Comment · Area_Chair_JSqg · 2025-08-05
> >
> > yFbK, please could you take a look at the author response above and whether it addresses any remaining concerns you have, e.g. the resource intensive method

---

> > > ### Comment · Area_Chair_JSqg · 2025-08-07
> > >
> > > Hi yFbK,
> > >
> > > As a reminder, the discussion period ends tomorrow on the 8th Aug AoE time and NeurIPS guidelines state that you must contribute to the discussion before signing off your review. Please could you take a look at the author's rebuttal above and write your thoughts.
> > >
> > > Thanks,
> > > AC

---

### Note · Authors · 2025-08-12

Dear NeurIPS 2025 AC, SAC, and PC,

We sincerely thank all reviewers for their constructive feedback and the AC for facilitating the discussion. In this paper, we propose MAPLE, the first framework to adapt DPO-style preference optimization to cross-modal embedding learning, and achieve current SoTA performance on fine-grained retrieval tasks.

Before the rebuttal, three of the four reviewers had already provided positive feedback. These three reviewers actively participated in the discussion, acknowledged that our rebuttal had addressed their main concerns, and updated their scores to 4 or 5, with consistently positive ratings in clarity, significance, and originality, reflecting a strong consensus in favor of acceptance. The remaining reviewer, who assigned a borderline-reject score with low confidence, did not participate in the discussion. Their primary concerns, which overlapped with those raised by other reviewers, were also addressed in our rebuttal through clarifications and additional ablation studies.

During the rebuttal, we further strengthened the work with additional robustness experiments:
- **Model size robustness**: Even small reward models (e.g., Qwen2-VL-2B, InternVL3-1B) yield consistent performance gains, showing MAPLE’s ability to leverage small reward models effectively.
- **Architecture robustness**: MAPLE is agnostic to reward model type; using InternVL/SAIL-VL reward models with a Qwen2 policy still yields clear improvements, demonstrating generalization across model families.
- **Hallucination robustness**: MAPLE maintains strong performance across reward models with varying hallucination rates, demonstrating robustness to noise from hallucination- or bias-related preference supervision.

We believe these results, together with the positive consensus among engaged reviewers, demonstrate that MAPLE offers a technically novel, empirically validated, and robust solution to the fine-grained retrieval task, making it a strong fit for the NeurIPS community. We will also open-source all the codes and datasets once accepted by the conference.

Best regards,
The Authors of Submission 20195

---

### Decision · Program_Chairs · 2025-09-17

**Decision:**

Accept (poster)

**Comment:**

This paper ended with 2 borderline accept scores, 1 accept score, and 1 borderline reject. Initially, reviewers liked the paper for its method being well motivated; the clarity of writing; and highly comprehensive experiments. However, they had some concerns regarding the resource intensive nature of the proposed method and lack of discussion regarding hallucinations/viability of different MLLMs which were both mentioned by multiple reviewers. The rebuttal attempted to address these key issues, with a large discussion over the online training stage only requiring a smaller model (compared to the offline captions) as well as new results comparing different MLLMs and showcasing that MAPLE was not reliant on a single MLLM. After the discussion, the majority of reviewers ended with a positive final rating with one reviewer recommending reject. The AC has looked at the arguments from both sides and agrees with the majority of reviewers recommending acceptance in this case: the carefully thought out experiments and discussion within the rebuttal provide evidence for the claims in the paper and answer the reviewers earlier questions. Accordingly the AC recommends acceptance for this paper.

The AC reminds the authors to make the changes they promised during the rebuttal and any changes specified as necessary by the reviewers for the camera ready version.